# Bacterial Contamination of Antiseptics, Disinfectants, and Hand Hygiene Products Used in Healthcare Settings in Low- and Middle-Income Countries—A Systematic Review

Palpouguini Lompo [1,2,3,*], Esenam Agbobli [4], Anne-Sophie Heroes [2,3], Bea Van den Poel [2], Vera Kühne [2], Cyprien M. Gutemberg Kpossou [5], Adama Zida [6], Halidou Tinto [1], Dissou Affolabi [4] and Jan Jacobs [2,3]

1   Institut de Recherche en Science de la Santé, Clinical Research Unit of Nanoro, Nanoro, Ouagadougou 11 BP: 218, Burkina Faso
2   Department of Clinical Sciences, Institute of Tropical Medicine, Nationalestraat 155, 2000 Antwerp, Belgium
3   Department of Microbiology, Immunology and Transplantation, KU Leuven, Naamsestraat 22 Box 5401, 3000 Leuven, Belgium
4   Centre National Hospitalier Universitaire Hubert Koutoukou Maga, Cotonou 01 BP: 386, Benin
5   Hôpital Saint Jean de Dieu de Boko, Parakou BP: 487, Benin
6   Centre Hospitalier Universitaire Yalgado Ouédraogo, Ouagadougou 03 BP: 7022, Burkina Faso
*   Correspondence: palpouguini.lompo@crun.bf; Tel.: +226-70300142

**Abstract:** We conducted a systematic review of healthcare-associated outbreaks and cross-sectional surveys related to the contamination of antiseptics, disinfectants, and hand hygiene products in healthcare settings in low- and middle-income countries (PROSPERO CRD42021266271). Risk of bias was assessed by selected items of the ORION and MICRO checklists. From 1977 onwards, 13 outbreaks and 25 cross-sectional surveys were found: 20 from Asia and 13 from Africa. Products most associated with outbreaks were water-based chlorhexidine, chlorhexidine-quaternary ammonium compound combinations (7/13), and liquid soap products (4/13). Enterobacterales (including multidrug-resistant *Enterobacter cloacae, Klebsiella pneumoniae*, and *Serratia marcescens*) and non-fermentative Gram-negative rods were found in 5 and 7 outbreaks and in 34.1% and 42.6% of 164 isolates, respectively, from cross-sectional surveys. Risk factors included preparation (place, utensils, or tap water high and incorrect dilutions), containers (reused, recycled, or inadequate reprocessing), and practices (topping-up or too long use). Potential biases were microbiological methods (neutralizers) and incomplete description of products' identity, selection, and denominators. External validity was compromised by low representativeness for remote rural settings and low-income countries in sub-Saharan Africa. Outstanding issues were water quality, biofilm control, field-adapted containers and reprocessing, in-country production, healthcare providers' practices, and the role of bar soap. A list of "best practices" to mitigate product contamination was compiled.

**Keywords:** antiseptics; bacterial contamination; best practices; cross-sectional; disinfectants; hand hygiene; low- and middle-income countries; outbreak

## 1. Introduction

Healthcare-associated infections have a serious burden worldwide. One in four sepsis cases treated in hospitals are healthcare-associated; they have a mortality rate of 24.4%, increasing by two to three times in cases of antimicrobial resistance [1]. In low- and middle-income countries (LMICs), healthcare-associated infections affect twice as many patients compared to high-income countries (HICs), i.e., 15% versus 7% of hospitalized patients, respectively; differences are even higher (up to 20 times) in intensive care units, particularly among neonates [1].

The healthcare environment is increasingly acknowledged as a reservoir of multidrug-resistant (MDR) bacteria [2,3]. From this environment, they may be transmitted by the

hands of healthcare providers, colonize the skin and mucous membranes of patients, and subsequently cause healthcare-associated infections. Hand hygiene is a key tool in preventing this transmission and is part of the core components that are listed among the World Health Organization's (WHO's) Minimum Requirements for Infection Prevention and Control (IPC) programs [4]. Environmental cleaning and decontamination processes by the use of low-level disinfectants reduce the bacterial load in a hospital environment and figure among the IPC core components as well [4]. In addition, antiseptics are applied on skin and mucosa in preparation for surgery and invasive procedures [4,5].

Although designed to inactivate microorganisms, antiseptics, disinfectants, and hand hygiene products (further referred to as AS, DI, and HH products) can be contaminated with bacteria and can be reservoirs and vehicles of healthcare-associated infections and outbreaks [5–7]. Inspired by anecdotal field observations in hospitals in sub-Saharan Africa [8], we conduct a review about the bacterial contamination of AS, DI, and HH products, which shows an underrepresentation of LMICs [9] despite the serious challenges and gaps in IPC in LMICs [1,10,11]. We, therefore, decide to describe the situations in HICs and LMICs separately; the review for HICs is published elsewhere [9].

The present systematic review differs from previous narrative reviews (dating from before 2007) in its extensive search, including grey literature, and its appraisal of reporting quality and of risk of bias. Further, it includes soap products and assesses the impact of contamination (numbers of patients affected and case–fatality ratios). It appraises the microbiological spectrum using updated nomenclature and the associated risk factors, as well as attribution and transmission. Finally, best practices to mitigate the contamination of AS, DI, and HH products are listed.

## 2. Research Questions of the Review

The research questions of this review are as follows:

1. What is the frequency and microbiological spectrum of the bacterial contamination of AS, DI, and HH products used in human healthcare in LMICs?
2. What are the risk factors for bacterial contamination?
3. What are best practices to mitigate the risk of bacterial contamination?

## 3. Materials and Methods

### 3.1. Review Protocol and Registration

We conducted this systematic literature review according to PRISMA guidelines [12], addressing articles reporting (pseudo)outbreak investigations caused by contaminated AS, DI, and HH products, as well as cross-sectional surveys assessing these products for contamination. We developed a study protocol describing the research questions, search strategy, data extraction, and interest and registered it in the PROSPERO register under CRD42021266271 [13] (Supplementary Document S1).

### 3.2. Terms and Definitions

This review focused on products listed on the WHO Model List of Essential Medicines [14], as well as on other basic products used in healthcare facilities in LMICs. For disinfectants, focus was put on low-level disinfectants used for environmental decontamination. Environmental cleaning and decontamination processes—for which low-level disinfectants are used [15,16]—reduce the bacterial load in a hospital environment and figure among the IPC core components as well [4]. Examples of low-level disinfectants are alcohol, sodium hypochlorite (chlorine), hydrogen peroxide, quaternary ammonium compounds, phenolics, etc. [15,16]. Antiseptics and disinfectants inactivate microorganisms or inhibit their growth; they are applied on skin and mucosa and on inanimate surfaces, respectively [6]. Most antiseptics and disinfectants are liquid products (mostly water-based or alcohol-based) and are available on-site in containers, some of which have a dispenser. Hand hygiene products include alcohol-based hand rubs and soaps. Soaps are detergents facilitating the removal of dirt; they can contain an antiseptic or not (plain soap) and are available as bar or

liquid formulations. Products may be procured as ready-to-use or as concentrated products, which are diluted to a working concentration, mostly in a healthcare facility's pharmacy. More extended definitions of AS, DI, and HH products are listed in a complementary review addressing AS, DI, and HH products in HICs [9].

In this review, generic product names were used. In cases where only the brand or technical names were used in the articles assessed, the corresponding generic name was looked up (if possible, by retrieving the Material Safety Data Sheet), and both the brand and generic names were mentioned. The terms "aqueous" and "tincture" were consistently replaced with "water-based" and "alcohol-based", respectively; "methylated spirit" was interpreted as "denatured alcohol". Country income was defined for each country according to the World Bank classification at the year of publication of the article [17].

The terms outbreak and pseudo-outbreak were adopted from the referred articles themselves [7]. Outbreaks included infected and colonized patients, with colonization defined as the presence of a pathogen without signs of infection. The term pseudo-outbreak referred to false-positive cultures of clinical specimens caused by contaminated products in the absence of patient colonization, infection, and exposure [18,19], such as in the case of blood culture contamination induced by contaminated antiseptics used for wiping the stoppers of blood culture bottles before inoculation [9]. Cross-sectional surveys referred to series of products sampled and cultured at a given time point and outside the context of an outbreak. Categories of products (antiseptics, disinfectants, and hand hygiene products (soaps)) were used as mentioned in the referred articles.

The taxonomic status of the bacteria was verified according to the List of Prokaryotic names with Standing in Nomenclature, and updated species names were retrieved [20]. After initial citing of the original (superseded) name, the updated genus and species names were further used in the manuscript. When possible, antibiotic susceptibility data of bacteria were assessed: acquired antibiotic resistance was assessed by comparison with the wild-type expected resistance phenotype [21] of the given species, and multidrug resistance was defined as acquired non-susceptibility to at least one agent in three or more antimicrobial categories [22].

### 3.3. Search Strategy

The literature search was based on the following criteria and considered publications in English, French, German, Portuguese, or Spanish. No starting date was defined; the search was updated until 31 May 2022. Inclusion criteria were original research studies addressing bacterial contamination of AS, DI, and HH products comprising (pseudo)outbreak investigations and cross-sectional surveys conducted in healthcare facilities in LMICs. Exclusion criteria were editorials and reviews; studies limited to molecular typing and experimental studies; studies in communities and veterinary healthcare; studies addressing exclusively bacterial contamination of water, sinks, or other sanitary equipment; and studies from HICs.

Databases searched were PubMed, Google scholar, Scopus, African Journal Online, and the Worldwide database for nosocomial outbreak [23]. The following search concepts detailed elsewhere [9] were used in PubMed: "antiseptics OR disinfectants OR soaps" AND "bacterial" AND "contamination OR growth" AND "nosocomial infection". The main keywords were adapted for the search in the 4 other databases. In addition, snowball screening by scanning the references' lists for relevant articles was used to find additional publications. Other libraries such as Hinari Research for Health/WHO and grey literature databases, including Master thesis repositories, were also searched [24]. The articles retrieved were screened (title and abstract) for eligibility by 2 independent reviewers (PL and BVP) using Rayyan systematic review screening software [25]. Any conflict was resolved by discussion with a third reviewer (JJ) and subsequent consensus.

To compile best practices, guidelines from international and national institutions and their source documents were explored. These institutions included the WHO [4,5,26], the United States of America Food and Drug Administration (U.S. FDA), and the U.S. Center

for Disease Control and Prevention [27,28]. Other professional and scientific associations included were the International Federation of Infection Control [29], non-governmental organizations, and other non-profit organizations (Médecins Sans Frontières [30] and the Johns Hopkins Program for International Education in Gynecology and Obstetrics [31]).

### 3.4. Data Extraction

The following general data were extracted: year of publication, country, contaminated product (name, type, and concentration), population and wards affected, and type of study ((pseudo)outbreak investigation or cross-sectional survey). For review question 1 (frequency and spectrum of bacterial contamination), the following data were extracted: environmental study and microbiology methods (sampling, culture, colony count, species identification, and antimicrobial susceptibility testing), contaminating index bacteria (genus, species, and antimicrobial susceptibility profile), proportions of contaminated products (in cross-sectional surveys), formulation and characteristics (dilution) of contaminated products, and relatedness between environmental and clinical isolates. In addition, epidemic investigation methods (outbreaks) and sample selection (cross-sectional surveys) were recorded.

For review question 2 (risk practices contributing to bacterial contamination), risk factors along products, containers, procedures, and practices as aggregated elsewhere [9] were considered. Factors were grouped as observations versus assumptions. Further, attribution and transmission (either demonstrated or hypothesized) were recorded. In addition to the strict content of the research questions, interventions to contain the outbreak or cease a product's contamination were also assessed. All data were extracted by one reviewer (PL) and checked by a second and third reviewer respectively (BVP and JJ). The approved complete extracted data were compiled in an Excel database (Microsoft, Redmond, WA, USA) (Supplementary Document S2).

### 3.5. Risk of Bias Assessment and Synthesis of Results

The risk of bias and quality of reporting of the outbreak investigations were assessed according in compliance with selected outcomes of the Outbreak Reports and Intervention studies Of Nosocomial infection (ORION) and the Microbiology Investigation Criteria for Reporting Objectively (MICRO) guidelines [32–34], complemented by the outcomes listed by Moffa et al. [35]. The ORION checklist provides criteria to appraise outbreak reports or intervention studies related to healthcare-associated infections [36]; the MICRO checklist provides criteria to assess the methodology of cumulative microbiology and antimicrobial resistance data in studies and reports [34].

This assessment was conducted by one reviewer (PL) and checked by a second reviewer (JJ). Outcomes were formulated and adapted to the setting of AS, DI, and HH products. For the cross-sectional results, a further selection of relevant outcomes was made.

Results for both outbreaks and cross-sectional studies were condensed in a color-coded overview table, with scores for different outcomes based on the categories (green, yellow, or red) listed in Supplementary Table S1; this was conducted by one author (PL) and checked by a second (JJ). In the Results and Discussion Section below, findings were compared with data from HICs, which were published in a separate review [9].

## 4. Results and Discussion

Supplementary Table S2 compiles the main findings of articles investigating outbreaks associated with contaminated AS, DI, and HH products. Supplementary Tables S3 and S4 compile the main findings for cross-sectional surveys assessing, respectively, antiseptics/disinfectants and soap products.

### 4.1. (Pseudo)outbreaks and Cross-Sectional Surveys: Overview

Figure 1 summarizes the search and screening process. Among the excluded articles with reason, some originated from LMICs: one article from India was excluded because it

was a purely experimental study [37]; three articles from Tunisia reported about a single outbreak by contaminated eosin (a chemical dye) as a reservoir [38–40]. In addition, two outbreak investigations found contaminated antiseptics but were excluded, as another fomite was identified as the source of the outbreaks [41,42] (Supplementary Table S5). One cross-sectional survey was removed from the analysis because the name and product category of the investigated product were not mentioned [43].

The final panel consisted of 38 articles, among which were 12 outbreaks, 1 pseudo-outbreak (together representing 34.2% of articles), and 25 (65.8%) cross-sectional surveys, versus 114 articles (71.1% (pseudo)outbreaks and 28.9% cross-sectional surveys) from HICs. Only a single article from LMICs reported a (pseudo)outbreak [44] compared to 13 in HICs. As there was only a single (pseudo)outbreak, the text further groups this article together with the outbreak reports, unless otherwise stated. The earliest publication dated from 1977 [45], exactly 20 years after the first publications from HICs [46]. During the 1980s and 1990s, publications from LMICs were rare, but from 2000 onwards, numbers increased and also comprised liquid soap products, in line with observations from HICs; the 2:1 rate of cross-sectional surveys versus (pseudo)outbreak articles remained stable over time (Figure 2) but was the opposite of what was observed in HICs [9].

A total of 26 (68.4%) out of 38 articles originated from middle-income countries, with only 2 outbreaks reported from low-income countries [47,48]. Over half of the articles (20/38, 52.6%) originated from Asia, particularly from India, Malaysia, and Thailand (Table 1). South America and the Caribbean accounted for 5 articles and Africa for 13 articles, 12 of which originated from sub-Saharan Africa, including 7 from Nigeria and only 1 from a French-speaking country. Central Africa and LMICs from Oceania were not represented [49]. For sub-Saharan Africa, the poor representation reflected the lack of bacteriology testing services and poor IPC in healthcare facilities: according to a recent survey, only 1.3% of 50.000 medical laboratories in 14 African countries conducted bacteriology testing [50], and only one-quarter (26%) of healthcare facilities in sub-Saharan Africa had basic environmental cleaning services in place [1]. There were no time or geographical clusters of publications among the outbreak reports and the cross-sectional surveys.

Out of 12 outbreak articles providing information, 11 occurred in an urban tertiary care setting. All outbreaks involved a single hospital; the affected wards comprised pediatrics (n = 7), surgery and related wards (n = 5), neonatology (n = 4), intensive care units, and hematology–oncology and dialysis (n = 3 each). The outbreak median (range) duration was 12 weeks (1 week–7.25 years); five outbreaks extended more than six months, of which four exceeded a one-year duration. The median (range) number of patients affected was 18 (5–361). In two outbreaks, colonized patients were reported [51,52]. Bloodstream infections were reported in 11/13 outbreaks, either alone (n = 4 outbreaks) or associated with specific foci, such as meningitis, wounds, and urinary tract infections (n = 7). These data were comparable to those reported from HICs (29 (1–151) patients affected and 11 (1–104) weeks duration), particularly when the *Burkholderia cepacia* outbreak associated with contaminated ethanol (411 bloodstream infection episodes in 361 patients and 7-year duration) [53] was subtracted from the comparison.

Ten articles reported patient outcome: the median (range) of the case–fatality ratio was 26.0% (0.0–88.5%), and the aggregated case–fatality ratio was 9.1%. This figure was significantly higher than that observed in HICs (median: 0.0% (0.0–60.0%); aggregated case–fatality ratio: 6.1%; $p = 0.027$, chi square). The median (range) number of deaths was four (from one to eight). In three outbreaks, case–fatality ratios were ≥40% [41,49,51]. Case–fatality ratios ≥ 20% were reported from five outbreaks in high-risk wards (surgery, neonatology (n = 2), pediatrics, and hemato-oncology) or were associated with interventions. The implicated organisms were MDR *Serratia marcescens, Elizabethkingia meningosepticum*, MDR *Klebsiella pneumoniae, Enterobacter cloacae*, and carbapenem-resistant MDR *Enterobacter cloacae*) [48,49,51,54,55]. In one outbreak (postsurgical infections associated with chlorhexidine gluconate (CHG) contaminated with *Achromobacter* spp.), a single death (among 59 affected patients) was considered unrelated to the infection [52]. When reported,

triggers pointing to a possible outbreak were an unusual high incidence of postoperative wound infections by *Pseudomonas aeruginosa* [47] and the occurrence of a previously rarely observed species, i.e., *Achromobacter denitrificans* [56].

Most (84.0%, 21/25) cross-sectional surveys aimed to assess the proportion of contamination of in-use products. Other surveys also aimed to study factors contributing to contamination [57–59] and to study the causative organisms, their resistance to antibiotics and antiseptics, and their genetic relatedness [60–62]. In two surveys, antiseptics and disinfectants were part of other fomites assessed (high-touch surfaces, cleaning tools, patient care items, and leftover vials of medicines) [63,64]. Three surveys assessing soaps compared the contamination between bar and liquid soap, respectively [65–67]. Routine infection control monitoring detected an anecdotal observation of contaminated in-use alcohol [68]. Seventeen surveys studied only one hospital. One survey (5 hospitals) was part of a larger study assessing IPC in public maternity units in Kenya [69], and a nationwide survey (addressing hospitals of all hierarchic levels (n = 39)) was performed in Thailand [58]. Three other surveys (in Malaysia and Nigeria) were comprised respectively of 6, 16, and 20 hospitals [59,63,70]. In addition to tertiary hospitals in urban settings (n = 17 articles), secondary hospitals (n = 8) and private facilities (n = 1) were addressed.

The median (range) number of different products tested per survey was 5 (1–10); the median number of samples tested (for 24/25 surveys providing data) was 94 (12–16,142 samples), which is higher than in HICs (median: 48 (1–492 samples) [9]. The nationwide surveys from Malaysia and Thailand each assessed more than 10,000 samples [58,70]. Compared to HICs, cross-sectional surveys also more frequently comprised general and private hospitals, as well as multicenter surveys. The large sample sizes allowed for making comparisons between wards and hospitals [57,58,62,63,70–72], different dilutions and formulations (alcohol- versus water-based products, bar versus liquid soaps) [57,58,62,65–67,69,70], and different storage conditions [57,63,73]. Products most frequently selected were phenol (12 surveys), chlorine (n = 10), alcohol (n = 9), chlorhexidine gluconate-quaternary ammonium compounds (CHG-QUAT), and liquid soaps (8 surveys each); phenol and chlorine were more frequently assessed compared to HIC surveys (6 and 1 surveys, respectively).

In addition to methodological issues (see below in Section 4.3), challenges were product names (see footnote of Supplementary Table S3), formulations, and origins. Some brand names (e.g., Lysol and Mercurochrome) stood for different generic products over time and in different countries, and the active components of the products used in the article were not retrievable. Likewise, products not or no longer marketed as antiseptics were listed (e.g., acriflavine, Mercurochrome (in its mercury-containing formulation) and Methimasol), and some products listed as disinfectants (e.g., Harpic and Biotex) were probably household-grade cleaning products. Further, some papers did not list products' concentrations [59].

As was the case for articles from HICs [9], the terminology of antiseptics versus disinfectants was not always correct; and example is povidone-iodine categorized as a disinfectant [70]. In some articles, the terms antiseptics and disinfectants were used interchangeably [58,60,74].

*4.2. Products Involved*

Products associated with outbreaks comprised water-based chlorhexidine gluconate (CHG) and chlorhexidine-quaternary ammonium compound (CHG-QUAT) combinations (representing half (7/13) of the articles), as well as ethanol and chlorine (1 product each) (Table 2). Almost half (n = 6) of these products were primarily used as skin antiseptics for intravenous catheter care [44,53,56] or for topical wound care [47,55,56]; one CHG product and a Dakin solution (i.e., a stabilized chlorine product used as an antiseptic) were used as disinfectants [49,75]. In one article, CHG was used both as a disinfectant (to soak nasal suction catheters) and for hand hygiene [55]. In addition, liquid soap products (among which were two antiseptic soaps) were reported in four articles [48,51,76,77].

Among 65 products found contaminated in cross-sectional surveys (Table 3), CHG, QUAT, and CHG-QUAT products (all but one water-based) represented 17 (26.2%) products,

and phenol compounds and chlorine accounted for 15 (23.1%) and 12 (18.5%), respectively. Liquid and bar soaps accounted for eight (12.3%) and four (6.2%) products, respectively. Alcohol products were reported in seven articles (10.8%) [58,63,64,68,70,74,78]. Products were mainly termed disinfectants (n = 51 articles). Indications for use (described in 13 articles) were disinfection of devices and instruments (forceps, thermometers, and suction and ventilation tubes (n = 6 surveys) [45,63,64,70,79,80], skin antisepsis (n = 2) [81,82], and hand hygiene (n = 6) [65–67,71,73]. In one report products, were used both for antisepsis and hand hygiene [64].

Concentrations of products assessed in outbreak articles were expressed either as dilutions or concentrations and varied: for water-based CHG, they ranged from a 1/2000 dilution of a 5% stock solution [47] to 2% and 4%, respectively (which are among the highest concentrations of products marketed) [44,52]. In one article, a 1/2000 dilution of a 5% CHG-QUAT solution was contaminated [75]. Expressed as number of samples contaminated versus total number assessed, cross-sectional surveys consistently showed that contamination of CHG, phenol, and chlorine was related to low concentrations (or high dilutions) and water-based formulations [57,70,79]. These observations were also made in articles from HICs [9]. Contamination was, however, also noted in alcohol-based CHG [70]. The high number of surveys (n = 7) showing contaminated ethanol was intriguing, but in two of these surveys, contamination was demonstrated by pre-enrichment of the samples. This is a procedure that overestimates the contamination (see below in Section 4.3) [64,74]. Further, in three other articles, cultures of alcohol and alcohol-based products (including CHG and iodine tinctures) remained negative [47,60,82].

The panel of contaminated products in outbreaks and cross-sectional surveys in LMICs was, overall, comparable to that observed in HICs, but some differences were noted, e.g., the proportion of chlorine and phenol products was higher in LMICs. As was the case for HICs, soap products were assessed from the 2000s onwards [9], and the proportion of soaps associated with outbreaks in LMICs was higher compared to HICs (4/13 (30.7%) versus 11/81 (13.5%), respectively) [9]. In LMICs, contamination was rarely observed for povidone-iodine, but this product was assessed in only three surveys [58,70,80]; further, an additional outbreak investigation from India (in which an alternative reservoir was identified) showed contamination of povidone-iodine with *Pseudomonas aeruginosa* [42] (Supplementary Table S5).

Multicenter surveys revealed substantial variation in contamination ratios per hospital [58,59,69,70,72,73]. A nationwide survey in Malaysia found product contamination rates per hospital ranging from 0.5% to 19.5% [70]; in one from Thailand, contamination ratios per hospital level were 0.0%, 0.7%, 3.3%, 2.3%, and 1.0% at the university, regional, provincial, district, and private levels, respectively [58]. One study in Nigeria found no large differences between product contamination ratios from different wards (overall ratio: 63.1%; range per ward: 50.0–72.2% [57]; another survey (assessing liquid and bar soap) found no relation between time of sampling (morning versus afternoon) and contamination ratio [62].

Among the four surveys that included both bar and liquid samples, three found higher contamination ratios among bar soap samples [65–67]; in two of them (assessing > 40 samples for each of bar and liquid samples), the proportions of contaminated bar soaps were 4-fold and 20-fold higher than in liquid soaps (30/50 versus 7/44 and 61/99 versus 2/60, respectively) (Supplementary Table S4) [66,67]. The remaining survey found 11.4% (44/378) of liquid soap products contaminated versus none among the bar soap samples, but the latter comprised only five samples [62]. Data allowing comparison between plain and antiseptic soaps were not available.

### 4.3. Epidemic and Microbiological Methods Used

Most (12/13) outbreak investigations conducted a clinical epidemic study to orient the environmental sampling, but half of them (n = 6) provided no details. The other seven investigations used case definitions and conducted case-control studies (5 and 2 investigations,

respectively). One investigation combined a cohort and a case-control study [53], and another sent a questionnaire to clinicians, together with a laboratory report [56]. In one investigation, AS, DI, and HH products were directly targeted (i.e., without a prior epidemic study) [47].

All but one environmental investigation (n = 12) assessed a wide range of fomites, such as liquids (e.g., milk for neonates, dialysate, and intravenous and irrigation solutions), high-touch surfaces (e.g., door handles, bedrails, trollies, and trays), and medical devices (e.g., catheters, tubing, endotracheal tubes, ultrasound scanners, stethoscopes, and infusion pumps). One investigation assessed the contamination of indoor air [55]. Upstream analyses along the chain of supply were conducted in four investigations and covered stock solutions and distilled water used for dilution in pharmacies [47,53–55]. Four investigations mentioned culturing unopened sealed bottles or sachets [44,51,52,77]. Four investigations assessed healthcare provider hands [54,76,77], patients (throat and rectal swabs) [51,55], and mothers of newborns (throat swabs [55]).

Most (72.0%, 18/25) cross-sectional surveys did not mention the method of sample selection. Three surveys used random selection [63,70,82], one survey selected products according to acceptability and frequency of use [80], and another (assessing soap products) selected samples from the sinks of toilets and working rooms [66]. One survey calculated a sample size targeted to the precision of the proportion of all fomites assessed [64], and another randomly selected 5% of all the products combined [70]. A nationwide study in Thailand enrolled hospitals (n = 39) according to representativity for hierarchic level and geographical localization [58].

All the surveys assessed in-use products. In addition, three, six, and one survey assessed freshly prepared products at pharmacies [58,70,81], stock samples in pharmacies and wards [57,59–61,81,82], and sealed original containers [71], respectively. In addition, one survey assessed boiled and tap water used for dilution [57]. Two-thirds (66.7%, 10/15) of surveys reporting information were conducted in multiple hospital wards, most frequently surgery and related wards (n = 10 surveys), pediatrics (n = 6), intensive care units (n = 5), obstetrics and gynecology (n = 4), and neonatology (n = 3).

Microbiological culture methods were detailed in 76.3% (29/38) of the articles (7 outbreaks and 22 cross-sectional surveys). Four articles used direct plating on agar media [47,58,66,81]. Seventeen articles used the Kelsey–Maurer method (or a modification of this method), which consists of a 1/10 dilution step of a product (to dilute the biocide effect) and subsequent plating on solid culture media [9,45,83]. Among these articles, 10 used a neutralizer to inactivate the biocide activity of the products, mostly 3% Tween 80. In seven articles, subculturing was performed in duplicate, with incubations at 37 °C and room temperature; incubation times varied considerably (between 24 h and 72 h at 35–37 °C and between 72 h and 7 days at room temperature). Two articles used a modified Kelsey–Maurer method but incubated the products diluted in broth from 24 h to 7 days instead of plating on culture media within 1 h after dilution [74,77]. Likewise, two articles inoculated and incubated the products in enrichment broths before subculturing (one article combined both methods) [51,64]. Culture techniques relying on broth enrichment cultures only (used in references [64,74,77]) can recover very low concentrations of organisms and, as such, overestimate the contamination rate, as AS, DI, and HH products are not sterile [9].

Quantitative cultures were performed by the pour plate method (n = 3) [53,59,61] or by inoculating 10-fold dilutions (n = 4) [60,71,72,84]. Two of these papers also used membrane filtration to assess viable counts [59,71]. Five articles mentioned the bacterial count at which the Kelsey–Maurer test was interpreted as positive, i.e., >250 and >1000 colony-forming units/mL (CFU/mL), respectively [45,57,63,70,79]. Seven articles reported high bacterial counts but did not provide details about the method used [45,59–61,65,71,75].

Culture media used for direct plating and subculturing were mostly blood agar and nutrient agar (13 and 11 articles, respectively) as general media; one study used Thayer Martin agar as a selective medium for *Elizabethkingia meningoseptica* [55], and another used cetrimide agar to detect *Burkholderia cepacia* [53]. Methods for isolate identification (for

33/38 articles describing these methods) were mostly conventional phenotypic testing (n = 26 articles; in some articles, complemented by commercial kits [53,66,70,77] and serotyping [55]. Other methods were automate-based phenotypic testing (n = 3) [44,56,68] and MALDI-TOF [67]. In 15 articles, antibiotic susceptibility testing (AST) was performed, with all but one by disk diffusion methods. Fourteen studies specified guidelines for interpretation; eight of them mentioned the version number or year of publication (for details, see Section 4.4).

Relatedness between index and environmental isolates was assessed by phenotypical methods (serotyping, pyocine typing, colony pigment, and AST results [44,47,53,55,76]) or molecular testing (plasmid testing, Pulsed-Field Gel Electrophoresis, PCR-based methods, Rep-PCR, RAPD, and whole-genome sequencing [44,48,51–54,56,75,77]) (Supplementary Table S2). Two cross-sectional surveys assessed relatedness between contaminating bacteria and bacteria isolated from clinical specimens by phenotypic identification and AST results [81] and Pulsed-Field Gel Electrophoresis [62].

As part of a root cause analysis, two articles used membrane filtration to assess water used for dilution [53,57], and one of them used sodium thiosulphate to neutralize chlorine [53].

As part of outbreak investigation, procedures (water treatment, cleaning, maintenance, dialysis, standard IPC procedures, and procedures for catheter care) were reviewed in six investigations [44,51,53,55,75,76], combined with interviews of staff [53,56,75]. Three cross-sectional surveys (from Trinidad and Tobago and Thailand) combined a microbiological survey with a questionnaire for staff to understand and map the lifecycle of products in a health facility [58,60,73]. Procedure reviews, interviews of staff, and observations of practices were part of other surveys [61,62,69–71,73,74,79].

In addition to the methodological limitations described in Section 4.1, none of the outbreak reports referred to the ORION guidelines [32,33], cross-sectional studies failed to describe sample selection, and there was a plethora of culture techniques. All of these observations are in line with findings from HICs [9]. In addition to these limitations, outbreak investigations from LMICs were well-performing in terms of clinical epidemic studies and samplings of high-touch, high-risk fomites. Compared with HICs, outbreak investigations in LMICs more frequently assessed staff and patients for colonization (conducted by only 9/68 outbreaks in HICs) but less frequently assessed the upstream tracks of the products (58.0% among 68 outbreak investigations in HICs) [9]. Among both outbreak investigations and cross-sectional surveys, many used state-of-the-art molecular testing and conducted in-depth interviews and observations to understand the root cause analysis of the contamination; some used advanced techniques (visualization of biofilm or whole-genome sequencing) [48,75] and addressed high sample sizes among multiple centers [58,70].

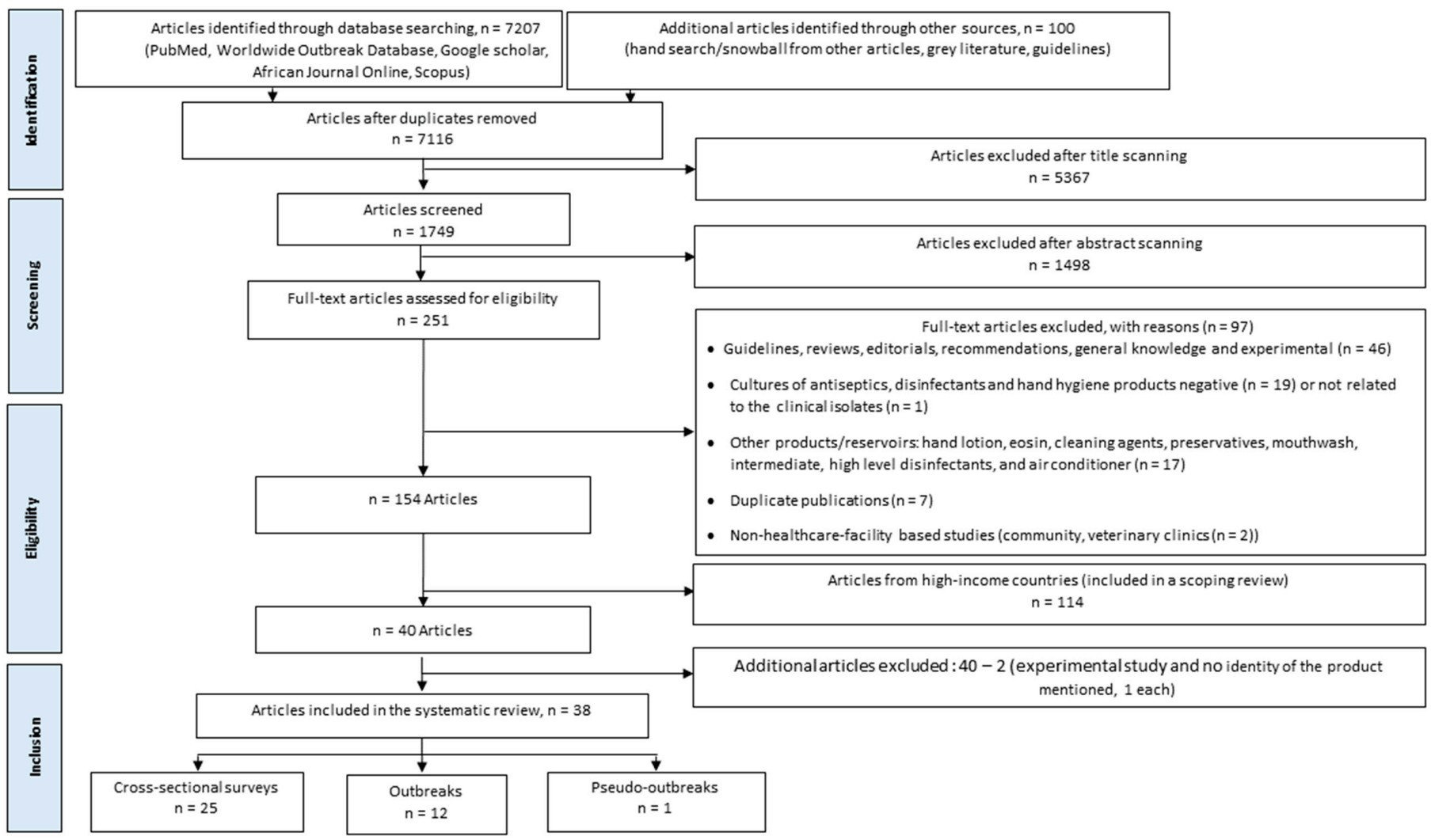

**Figure 1.** Flow chart of literature search for bacterial contamination of antiseptics, disinfectants, and hand hygiene products in low- and middle-income countries.

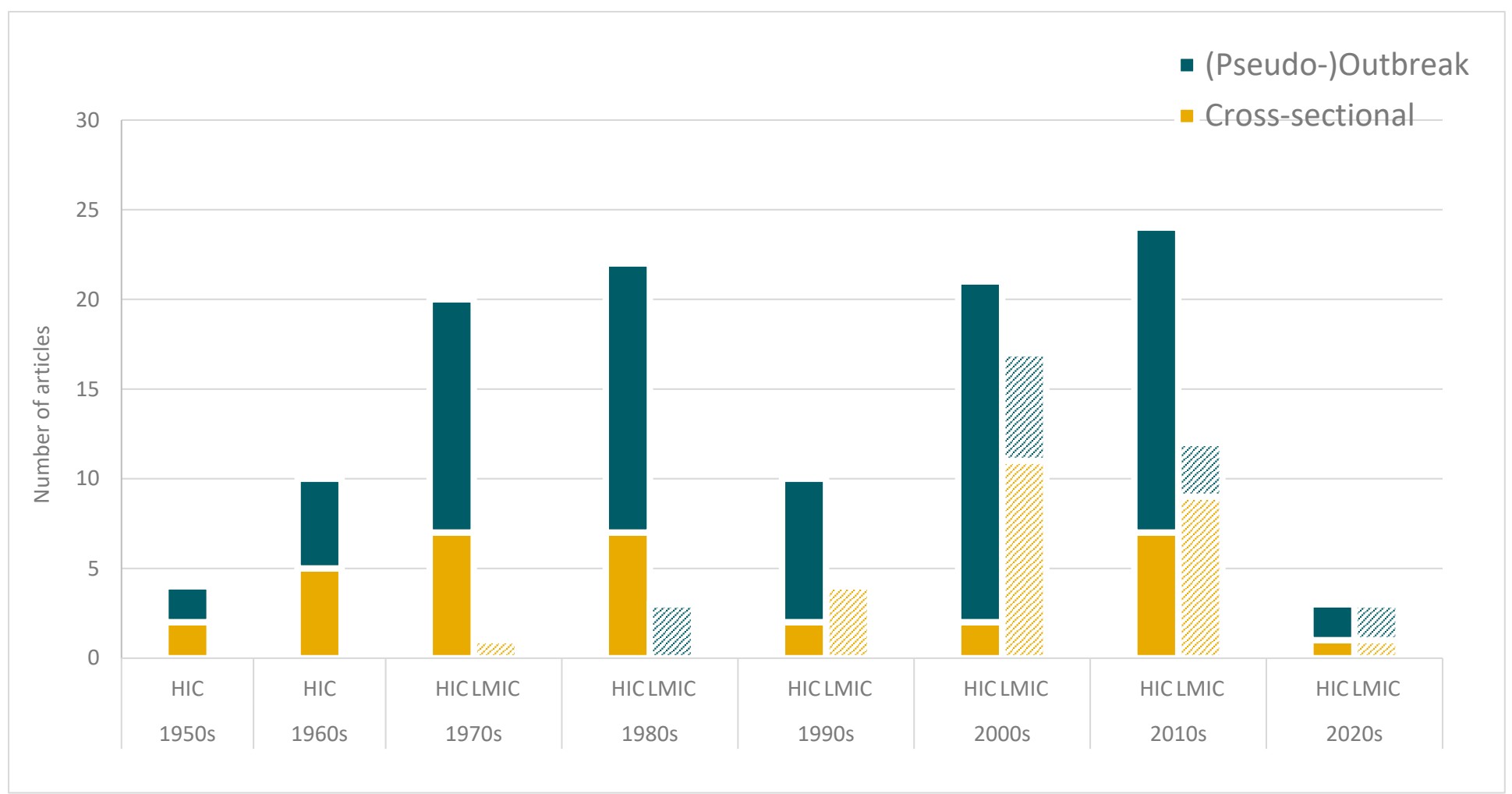

**Figure 2.** Distribution of (pseudo)outbreak reports and cross-sectional surveys investigating bacterial contamination of antiseptics, disinfectants, and hand hygiene products in high, low-, and middle-income countries among decades. Solid, filled bars represent high-income countries (HICs), while dashed bars refer to low- and middle-income countries (LMICs).

**Table 1.** Geographic distribution of articles reporting bacterial contamination of antiseptics, disinfectants, and hand hygiene products in healthcare facilities in low- and middle-income countries. The numbers represent articles; countries are categorized according to the United Nations geoscheme [85].

| United Nations Geoscheme/Countries | Cross-Sectional | Outbreak [a] | Total |
|---|---|---|---|
| Eastern Africa | 3 | - | 3 |
| Ethiopia | 2 | - | 2 |
| Kenya | 1 | - | 1 |
| Northern Africa | - | 1 | 1 |
| Tunisia | - | 1 | 1 |
| Southern Africa | - | 1 | 1 |
| South Africa | - | 1 | 1 |
| Western Africa | 6 | 2 | 8 |
| Nigeria | 6 | 1 | 7 |
| Senegal | - | 1 | 1 |
| Eastern Asia | 1 | - | 1 |
| China | 1 | - | 1 |
| Southeastern Asia | 5 | 3 | 8 |
| Malaysia | 2 | 2 | 4 |
| Thailand | 3 | 1 | 4 |
| Southern Asia | 4 | 2 | 6 |
| India | 4 | 1 | 5 |
| Nepal | - | 1 | 1 |
| Western Asia | 4 | 1 | 5 |
| Iraq | 1 | - | 1 |
| Lebanon | - | 1 | 1 |
| Palestine | 2 | - | 2 |
| Turkey | 1 | - | 1 |
| Latin America and the Caribbean | 2 | 3 | 5 |
| Argentina | - | 1 | 1 |
| Brazil | 1 | - | 1 |
| Colombia | - | 1 [a] | 1 |
| Mexico | - | 1 | 1 |
| Trinidad and Tobago | 1 | - | 1 |
| Total | 25 | 13 | 38 |

[a] Including a (pseudo)outbreak [44].

**Table 2.** Products implicated in outbreaks and (pseudo)outbreaks (n = 13) associated with contaminated antiseptics, disinfectants, and hand hygiene products in healthcare facilities in low- and middle-income countries. Numbers represent the number of articles. Abbreviations: CHG = chlorhexidine gluconate; QUAT = quaternary ammonium compounds.

| Decades | Alcohol | CHG [a] | CHG-QUAT [b] | Chlorine | Liquid Soap [c] | Total |
|---|---|---|---|---|---|---|
| 1980s | - | 2 | - | 1 | - | 3 |
| 2000s | 1 | 1 | 2 | - | 1 | 5 |
| 2010s | - | 1 [a] | - | - | 2 | 3 |
| 2020s | - | 1 | - | - | 1 | 2 |
| Total | 1 | 5 | 2 | 1 | 4 | 13 |

[a] Including one (pseudo)outbreak. [b] In one outbreak reporting contamination of CHG-QUAT, the product was used as a disinfectant, as well as a hand hygiene product; the latter was implicated in handborne bacterial transmission. [c] Including antiseptic soap (CHG soap, n = 1), plain soap (n = 1), and no information on antiseptic or plain soap (n = 2).

**Table 3.** Products implicated in cross-sectional surveys (n = 25) reporting contaminated products of antiseptics, disinfectants, and hand hygiene products in healthcare facilities in low- and middle-income countries. Numbers represent contaminated products per survey; numbers exceed the number of articles since several articles reported more than one contaminated product. CHG = chlorhexidine gluconate; $H_2O_2$ = Hydrogen peroxide; QUAT = quaternary ammonium compound.

| Decades | Alcohol | CHG [a] | CHG-QUAT | QUAT | Iodophor | Phenol [b] | Chlorine | $H_2O_2$ | Liquid Soap [c] | Bar Soap [d] | Total |
|---|---|---|---|---|---|---|---|---|---|---|---|
| 1970s | - | - | - | - | - | 1 | - | - | - | - | 1 |
| 1990s | 2 | 1 | 2 | 1 | 1 | 5 | 3 | - | - | - | 15 |
| 2000s | 2 | 4 | 5 | 2 | - | 7 | 6 | 1 | 2 | 3 | 32 |
| 2010s | 2 | - | 2 | - | - | 2 | 3 | - | 6 | 1 | 16 |
| 2020s | 1 | - | - | - | - | - | - | - | - | - | 1 |
| Total | 7 | 5 | 9 | 3 | 1 | 15 | 12 | 1 | 8 | 4 | 65 |

[a] Including water-based CHG (n = 4) and alcohol-based CHG (n = 1). [b] Phenol compounds included phenol (n = 10), PCMX (para-chloro-meta-xylenol, n = 4), and DCMX (dichloro-meta-xylenol, n = 1). [c] Liquid soap included antiseptic soap (n = 2; one soap with CHG and, for the other, no product name was mentioned) and plain soap (n = 3); for the other products, no information was provided. [d] Bar soap included plain soap (n = 2); for the other products no information was provided.

*4.4. Microorganisms Involved*

A total of 18 isolates were recovered from 13 outbreak-related articles (Table 4). One article retrieved two Enterobacterales species from a soap product, while another obtained five species of coagulase-negative staphylococci from contaminated QUAT-CHG [51,77]. Enterobacterales were associated with five outbreaks, with three of them related to liquid soap products [48,51,76]. In the two other outbreaks, *Enterobacter cloacae* was associated with a Dakin solution (a chlorine product) [49], and *Serratia marcescens* was obtained from a CHG solution [54]. Non-fermentative Gram-negative rods accounted for seven outbreaks, six of which were obtained from CHG or CHG-QUAT products [44,47,52,55,56,75]. *Burkholderia cepacia* accounted for three outbreaks, followed by *Achromobacter* spp. (n = 2), *Pseudomonas aeruginosa*, and *Elizabethkingia meningoseptica* (one outbreak each). A single alcohol product (ethanol 70%) was contaminated by *Burkholderia cepacia* [53].

Among a total of 164 single species obtained in 25 cross-sectional surveys (Table 5), Enterobacterales and non-fermentative Gram-negative rods represented 34.1% and 42.6% of isolates, respectively; the remaining species were Gram-positive cocci and rods, including *Staphylococcus aureus* and *Bacillus* spp., respectively. Among the Enterobacterales, *Enterobacter* spp. ranked first (17, 10.3% of all isolates), followed by *Escherichia coli* and *Klebsiella* spp. (11 and 10 isolates, respectively). Among the Gram-negative non-fermentative bacteria, *Pseudomonas aeruginosa* was the most frequent (retrieved in 20 surveys).

The spectrum of bacteria involved in outbreaks and detected as contaminating flora in cross-sectional surveys is in line with the findings from HICs [9]. Notable differences were the high proportion of Gram-positive bacteria and the low frequency of *Burkholderia*

*cepacia* in cross-sectional surveys in LMICs. However, the high proportion of Gram-positive bacteria may be biased, as five non-aureus staphylococci species were reported from a single outbreak investigation that exclusively used enrichment broths [77]. Likewise, one survey reporting two Gram-positive isolates cultured the caps (stoppers) of containers and leftover samples [64]. Further, using numbers of surveys that recovered particular species is not necessarily a reliable proxy for frequency. For example, in the single cross-sectional survey mentioning *Burkholderia cepacia*, this species represented the most frequently cultured organism [71].

For a brief overview of the nature, habitat, and clinical significance of the non-fermentative Gram-negative rods implicated in healthcare-associated outbreaks, we refer to the complementing review [9]. Of note, some of these bacteria are intrinsically resistant to CHG and QUAT and produce biofilms protecting bacteria from biocides (see Section 4.5). *Elizabethkingia meningoseptica* (formerly *Flavobacterium meningosepticum* and *Chryseobacterium meningosepticum*) shares the ability of non-fermentative Gram-negative rods to thrive in humid environments but stands out for invasiveness and high case–fatality ratio [86]. A large-scale outbreak caused by *Burkholderia cepacia* in 70% ethanol was, in part, ascribed to its nutritional versatility, i.e., metabolization of alcohol [53]. Enterobacterales are common hospital-associated bacteria: as an example, *Klebsiella pneumoniae* (responsible for an outbreak related to contaminated soap containers in a pediatric oncology unit) represented 20% of blood cultures isolates in a ward [51]. In such settings, suspicion of an outbreak can be easily overlooked [87].

Articles assessing bacterial counts above the threshold of the Kelsey–Maurer method reported a wide range of counts, of which the maximum exceeded $10^5$ CFU/mL (up to $10^7$ CFU/mL) in CHG, CHG-QUAT, para-chloro-meta-xylenol (PCMX), and chlorine- and phenol-based products [45,61,75], as well as in bar and liquid soaps [65,71]. These high counts are in line with those obtained from HICs [9]. In one paper, colony counting provided clues to the reservoir of an outbreak caused by *Burkholderia cepacia* in a dialysis unit: the dialysate showed low (30 CFU/mL) bacterial counts, whereas the QUAT disinfectant (later confirmed as the reservoir by molecular testing) showed very high counts ($10^5$ CFU/mL) [75].

Of the 15 articles that performed AST, some used an incomplete or inappropriate panel of antibiotics [61,74], did not display complete results [57,60,73,81], or listed uninterpretable or inconsistent results (no denominator, antimicrobial resistance pattern not compatible with species resistance phenotype) [61,73,76]. The remaining panel of interpretable bacteria–antibiotics combinations comprised several bacteria classified as MDR, including third-generation cephalosporin-resistant *Enterobacter cloacae* (one carbapenem-resistant) [48,49], *Klebsiella pneumoniae* [51], and *Serratia marcescens* [54]. The frequency of MDR bacteria was higher than that reported from HICs [9], and the above bacteria figure on the WHO list of pathogens prioritized for research [88]. Of note, some of the non-fermentative Gram-negative bacteria described displayed wild-type antibiotic resistance and did not fulfill the definition of MDR, but their wild-type (i.e., natural) antibiotic resistance by itself entailed resistance to multiple classes of antibiotics [21] and jeopardized effective antibiotic treatment with antibiotics locally available in LMICs. Examples were *Elizabethkingia meningoseptica* [55] and *Burkholderia cepacia* [44,53,75], both of which were resistant to third-generation cephalosporins. Additionally, *Elizabethkingia meningoseptica* was resistant to carbapenem antibiotics and the *Burkholderia cepacia* complex to aminoglycosides [21,86].

**Table 4.** Bacteria contaminating antiseptics, disinfectants, and hand hygiene products listed in 12 outbreaks and 1 (pseudo)outbreak in healthcare facilities in low- and middle-income countries. Numbers in cells represent articles citing bacterial species; numbers exceed the number of articles since one outbreak reported more than one bacterial species [77]. Details about the isolates (bacterial load and references) can be found in Supplementary Table S2. Abbreviations: CHG = chlorhexidine gluconate; CHG-QUAT = chlorhexidine-quaternary ammonium compound.

| Contaminating Bacteria | Alcohol | CHG | CHG-QUAT | Chlorine | Liquid Soap [a] | Total |
|---|---|---|---|---|---|---|
| Enterobacterales | - | 1 | - | 1 | 0/2/2 | 6 |
| *Enterobacter cloacae* | - | - | - | 1 | 1 | 2 |
| *Serratia marcescens* | - | 1 | - | - | 1 | 2 |
| *Citrobacter* spp. | - | - | - | - | 1 [e] | 1 |
| *Klebsiella pneumoniae* | - | - | - | - | 1 [e] | 1 |
| Non-fermentative Gram-negative rods | 1 | 4 | 1 | - | 1/0/0 | 7 |
| *Burkholderia cepacia* | 1 | 1 [b] | 1 | - | - | 3 |
| *Achromobacter* spp. [c] | - | 1 | - | - | 1 | 2 |
| *Pseudomonas aeruginosa* | - | 1 | - | - | - | 1 |
| *Elizabethkingia meningoseptica* | - | 1 | - | - | - | 1 |
| Gram-positive cocci [d] | - | - | 5 | - | - | 5 [d] |
| Coagulase-negative staphylococci | - | - | 5 | - | - | 5 |
| Total | 1 | 5 | 6 | 1 | 1/2/2 | 18 |

[a] Including antiseptic soap (n = 1), plain soap (n = 2), and no information provided (n = 2). [b] *Burkholderia cepacia* was reported in a (pseudo)outbreak. [c] *Achromobacter* species include *Achromobacter denitrificans* and *Achromobacter* spp. (n = 1 each). [d] Gram-positive cocci isolated only by enrichment (n = 5 isolates) reported in one survey [71], including *Staphylococcus haemolyticus*, *Staphylococcus epidermidis*, *Staphylococcus capitis*, *Staphylococcus saprophyticus*, and *Staphylococcus hominis* (n = 1 isolate each). [e] In one outbreak [43], two isolates were retrieved in the contaminated product.

*4.5. Factors Associated with Contamination*

Apart from the above documented factors, many articles discussed general or assumed risk factors beyond the articles' findings. Most of the factors described above largely overlap with those reported from HICs [9], apart from the absence of organic materials (gauzes and cotton balls) and biofilm in the present articles. Organic materials such as cork, gauzes, and cotton balls are well-known to inactivate CHG products [89,90]. Biofilms (i.e., bacteria attached to a surface within a polymeric substance) shield bacteria from toxic substances and typically develop in storage containers at rims, dispenser outlets, grooves, and surface abrasions [91–93]. Only three cases of intrinsic contamination were reported, but this number is in proportion with those reported from HICs. Human factors also played a role in practices conducive to contamination, such as topping-up, which was shown to be persistent in HICs, too [9]. Some of these practices are difficult to eradicate: in a follow-up paper, a team that conducted a multicenter survey in Malaysia [70] reported increased awareness and improved practices after an educational intervention but also noted the persistence of inappropriate "institutionalized" practices [94].

**Table 5.** Bacteria contaminating antiseptics, disinfectants, and hand hygiene products in healthcare facilities in low- and middle-income countries for a total of 26 cross-sectional surveys that provided detail about both products and bacteria. Numbers represent articles citing bacterial species; total numbers exceed the number of surveys since surveys detected more than one species. Details about the isolates (bacterial load and references) can be found in Supplementary Tables S3 and S4. Abbreviations: CHG = chlorhexidine gluconate; DCMX = dichloro-meta-xylenol; H₂O₂ = hydrogen peroxide; QUAT = quaternary ammonium compounds; PCMX = para-chloro-meta-xylenol. Subcategories: CHG: water-based/alcohol-based; liquid soap: antiseptic/plain/no information; bar soap: plain/no information.

| Contaminating Bacteria | Alcohol | CHG [a] | CHG-QUAT | QUAT | Iodophor | Phenol | Chlorine | H$_2$O$_2$ | Liquid Soap | Bar Soap | Total |
|---|---|---|---|---|---|---|---|---|---|---|---|
| Enterobacterales and non-cholerae Vibrio | 3 | 6/2 | 4 | 1 | - | 5 | 8 | 3 | 1/10/4 | 6/3 | 56 |
| *Enterobacter* spp. [b] | 1 | 4 | 1 | - | - | 1 | 3 | 2 | 3 | 2 | 17 |
| *Escherichia coli* | - | 1 | 1 | - | - | 3 | 1 | - | 3 | 2 | 11 |
| *Klebsiella* spp. [c] | 1 | - | - | 1 | - | 1 | 2 | - | 4 | 1 | 10 |
| *Proteus* spp. [d] | - | 1 | 2 | - | - | - | 2 | 1 | 2 | 1 | 9 |
| *Hafnia alvei* | 1 | 2 | - | - | - | - | - | - | - | - | 3 |
| *Serratia marcescens* | - | - | - | - | - | - | - | - | 1 | 1 | 2 |
| *Citrobacter* spp. | - | - | - | - | - | - | - | - | 1 | 1 | 2 |
| *Vibrio shilonii* | - | - | - | - | - | - | - | - | - | 1 | 1 |
| *Coliform* (not identified) | - | - | - | - | - | - | - | - | 1 | - | 1 |
| Non-fermentative Gram-negative rods | 3 | 7/5 | 7 | 4 | 1 | 9 | 8 | 1 | 1/6/2 | 6/10 | 70 |
| *Pseudomonas aeruginosa* | 1 | 1 | 2 | 1 | 1 | 2 | 4 | 1 | 3 | 4 | 20 |
| *Pseudomonas* spp. [e] | 1 | 3 | 1 | 1 | - | 1 | 2 | - | 5 | 2 | 16 |
| *Acinetobacter* spp. [f] | 1 | 2 | 2 | - | - | 2 | 2 | - | - | 3 | 12 |
| *Moraxella* spp. | - | 2 | - | 1 | - | 2 | - | - | - | - | 5 |
| *Achromobacter* spp. [g] | - | 2 | 1 | - | - | 1 | - | - | - | - | 4 |
| *Flavobacterium* spp. | - | 2 | - | - | - | 1 | - | - | - | - | 3 |
| *Burkholderia cepacia* | - | - | - | - | - | - | - | - | 1 | - | 1 |
| *Chryseobacterium indologenes* | - | - | - | - | - | - | - | - | - | 1 | 1 |
| *Stenotrophomonas maltophilia* | - | - | 1 | - | - | - | - | - | - | - | 1 |
| Other non-fermentative Gram-negative rods [h] | - | - | - | 1 | - | - | - | - | - | 6 | 7 |
| Other Gram-negative rods (not identified) | - | - | 1 | - | - | - | - | - | - | - | 1 |
| Gram-positive bacteria | 3 | 3/1 | 3 | 1 | - | 3 | 5 | 2 | 2/0/5 | 2/4 | 34 |
| *Bacillus* spp. [i] | 2 | 3 | 1 | - | - | 1 | 2 | 1 | 2 | 1 | 13 |
| *Staphylococcus aureus* | - | - | 1 | - | - | 2 | 2 | - | 2 | 4 | 11 |
| Coagulase-negative staphylococci | 1 [j] | 1 | - | 1 | - | - | 1 | 1 | 2 | - | 7 |
| Gram-positive rods (not identified) | - | - | 1 | - | - | - | - | - | - | - | 1 |
| *Corynebacterium* spp. | - | - | - | - | - | - | - | - | - | 1 | 1 |
| *Enterococcus* spp. | - | - | – | - | - | - | - | - | 1 | - | 1 |
| Yeast | - | - | - | - | - | - | - | - | 1 | 2 | 3 |
| Total | 9 | 16/8 | 15 | 6 | 1 | 17 | 21 | 6 | 4/16/12 | 15/18 | 164 |

[a] Including alcohol-based and water-based CHG. [b] Including *Enterobacter cloacae* (n = 4), *Enterobacter* spp. (n = 8), and *Enterobacter (Klebsiella) aerogenes* (n = 5). [c] Including *Klebsiella pneumoniae* (n = 4), *Klebsiella oxytoca* (n = 1), and *Klebsiella* spp. (n = 2). Two isolates of *Klebsiella pneumoniae* and one isolate of *Klebsiella pneumoniae* subsp. *ozaenae* were obtained by enrichment only [69]. [d] Including *Proteus mirabilis* (n = 7) and *Proteus penneri* (n = 2). [e] Including *Pseudomonas* spp. (n = 10), *P. oryzihabitans* (n = 2), *P. luteola*, *P. mendocina*, *P. oleovorans*, and *P. putida* (n = 1 each). [f] Including *Acinetobacter baumannii* (n = 2, of which one was recovered from alcohol by enrichment method [56]), *A. anitratus* (n = 2), *A. calcoaceticus*, *A. lwoffii*, *A. haemolyticus* (n = 1 each), and *Acinetobacter* spp. (n = 5). [g] Including *Achromobacter xylosoxidans* (n = 1) and *Alcaligenes* spp. (n = 3). [h] Including *Alishewanella fetalis*, *Arthrobacter* spp., *Empedobacter brevis*, *Halomonas aquamarina*, *Nesterenkonia lacusekhoensis* (one isolate each), and not identified (n = 2). [i] *Bacillus* spp. (n = 12; one isolate from alcohol by enrichment method [56]) and *Bacillus amyloliquefaciens* (n = 1). [j] one isolate of coagulase-negative staphylococci was isolated from enrichment method [56].

*4.6. Attribution and Transmission*

Among the outbreak investigations, attributions of AS, DI, and HH products as reservoirs were mainly provided by culturing of an index organism from a suspected product and demonstrating its identity with clinical isolates. In 8/13 investigations, identity

was demonstrated by phenotypic [47,49,76] or by genotypic methods [44,52,53,75,77]. In five outbreaks, identity was probable but not ascertained given inconsistent results among techniques used (e.g., same PFGE patterns but different AST profiles) or given co-occurrence of different phenotypes or genotypes [48,51,54,56,77]. Genetic diversity of healthcare-associated bacteria is not unusual and depends on the evolutionary rate (for *Pseudomonas aeruginosa*, this is high) [95]. Additional evidence was provided by time-relatedness: the onset of one outbreak coincided with the use of a water-based CHG [47], and in seven outbreaks, cases diminished or stopped after interventions [44,51–53,56,75,77].

In cases of intrinsic contamination, it was plausible that the product itself was the primary source of contamination [44,52,77], but in other outbreaks, it was not always clear whether the contaminated product was the unique reservoir and responsible for transmission. In one outbreak, the ubiquitous presence of the index organisms (*Elizabethkingia meningoseptica*) in other fomites and in the air precluded the authors from defining the contaminated CHG containers as the definite reservoir, although contaminated CHG containers were proportionally most frequently affected [55]. Likewise, in two other outbreaks that did not assess other fomites in addition to a liquid soap product, the latter was considered as a potential but not a definitely proven reservoir [48,77].

Transmission routes from reservoir to patients were discussed in 11/13 investigations. Direct contact was plausible when CHG was used as an antiseptic during central venous and urinary catheter insertion, pre-operative skin antisepsis, and wound care. In neonates, transmission was assumed to occur by direct invasion through the respiratory tract and abraded skin [44,47,52,55,56]. *Burkholderia cepacia* in contaminated 70% ethanol was assumed to be transmitted through intravascular catheter insertion by both skin antisepsis and disinfection of the rubber stoppers of heparin vials [53]. Products used as disinfectants spread by contact with (semi)critical devices, such as a transferring forceps (used to transfer gauzes during the insertion of intravascular catheters) standing in a jar filled with contaminated CHG-QUAT or soaking intravascular catheters in a Dakin solution [49,75]. Handborne transmission (i.e., transmission by contamination of healthcare workers' hands and, subsequently, patients) was assumed in three outbreaks associated with contaminated liquid soap [51,76,77].

Cultures of patients and staff sampled as part of outbreak investigations revealed different results. In the above-mentioned outbreak of *Elizabethkingia meningoseptica* in a neonate unit, 8.7%, 4.1%, and 11.0% of neonates, mothers, and patients from other wards had growth of the index organisms from the upper respiratory tract [55]. The hands of nurses assessed for index organisms grew *Staphylococcus haemolyticus* (2/12 cases) and *Serratia marcescens* (1/41 cases) [54,77] and were negative in another investigation [76]. Assessing healthcare workers for colonization by bacteria is not recommended, except in cases of specific bacteria and diseases or when specifically oriented by epidemiological investigation [27,96]. In addition, the implicated organisms had variable ability for hand colonization (high for Enterobacterales but low for non-fermentative Gram-negative bacteria), and the colonizing flora were transient and eliminated by hand hygiene [96]. For the culturing of patients, see Section 4.7.

Four cross-sectional surveys compared environmental isolates with clinical isolates from healthcare associated infections. Two of them found similarities in species identification and AST profiles [57,81], while the two others did not [62,73].

Attributions of reservoirs and elucidation of transmission routes among the outbreaks in LMICs are in line with the findings obtained from HICs [9]. Factors hampering reservoir and transmission investigations in HICs, such as availability of enough clinical isolates and products in use reported from HICs, were not mentioned in the present articles but are probably highly relevant to LMICs, given the low volumes of samples processed [8,97].

### 4.7. Interventions

Outbreak control interventions were reported in 11/13 investigations. In addition to removal of the contaminated products, replacements included substitution of contaminated

water-based CHG by alcohol-based CHG [47] or individually packed alcohol pads [53] and replacing liquid soap with another soap product or an alcohol-based handrub [76,77]). In the two outbreaks associated with intrinsically contaminated CHG (in Columbia and Argentina), the manufacturer and national regulatory authorities were notified [44,52].

Outbreak control further focused on the use of safe water (either boiled or distilled) [47,53,55] and the efficacy of containers' reprocessing steps (sterilization or soaking with chlorine) [51,56]. Elbow-commanded and smaller-volume containers were each implemented in one article [51,55]. Further, the practice of transfer forceps standing in CHG-QUAT-filled jars was banned [75], and two articles reinforced hand hygiene practices [55,76]. Other interventions included heat sterilization of prepared CHG solutions [55], "fogging" with stabilized hydrogen peroxide [48], and contact precautions and the cohorting or isolation of colonized and infected patients (in one article, combined with temporary unit closure) [44,51].

Follow-up after interventions was mentioned by seven investigations; the period of follow-up (provided by four articles) ranged from 4 weeks to 18 months. One article did not provide the results of the follow-up period [55], four articles reported an absence of new cases [51,75,77], and two others reported a considerable but incomplete reduction in cases [47,53].

Control measures applied to outbreaks associated with AS, DI, and HH products in LMICs are similar to those reported from HICs [9], apart from minor differences. Unlike the case for reports from HICs (n = 6), articles from LMICs less frequently mentioned training and education of staff (although this was obviously a risk factor for contamination) and more frequently addressed a safe water source. As was the case for articles from HICs, neither the design of the outbreak investigation nor the applied interventions allowed assessing the efficacy of the investigation [9]. However, not all interventions were evidence-based or proved to be effective: heat sterilization of CHG [55] is not possible, as it degrades the product [98], and $H_2O_2$ fogging reduced but did not eliminate infections associated with a contaminated soap dispenser [48]. Lack of access (related to stock rupture and financial barriers) precluded the consistent use of single-packed alcohol pads, reverting staff to the in-house production of 70% ethanol and subsequent re-occurrence of infections [53].

Two cross-sectionals survey organized educational activities for staff [62,94]. In another survey of liquid soap samples, a team installed a cleaning program (with disassembly of dispensers), introduced alcohol-based handrub in risk areas, and organized microbiology control at the reception of products [71].

All but one surveys further provided recommendations, mostly in line with the above discussed interventions: need for hospital policy, guidelines, and procedures (n = 3); appropriate reprocessing, including sterilization of containers (n = 3); monitoring products for contamination (n = 6); appropriate preparation performed by trained and competent staff (n = 6); and small-volume containers with short storage and in-use duration [78]. Other recommendations were the need for sterile products [68,77], monitoring concentrations of products [79], and the preference of liquid over bar soap, the latter expressed by three surveys that compared both types of soap [65,66].

For contact precautions, the WHO recommends actively screening asymptomatic colonization with carbapenem-resistant Enterobacterales and isolating or cohorting patients colonized or infected with carbapenem-resistant organisms [88]. In the case of carbapenem-resistant *Pseudomonas aeruginosa* and *Acinetobacter* spp., which are less effective colonizers compared to Enterobacterales, active screening for colonization depends on the local risk and setting [99]. The WHO recommends that screening and isolation or cohorting should mitigate potential harm, as well as negative social and psychological consequences, and, particularly in low-resource settings, be balanced against local prevalence, availability of resources, other IPC needs, and cultural perceptions of offensiveness and stigma [88].

*4.8. Internal and External Validity*

Internal validity: A color-coded bias score for relevant items of outbreak reports and cross-sectional surveys is presented in Tables 6 and 7. They depict a side-by-side view of the individual articles assessed according to the topics discussed above. Overall, outbreak investigations provided satisfactory-to-good information about outbreak setting, course, and index bacteria. Occasional poor scores concerned the description, terminology, and use of a contaminated product. The clinical epidemic and environmental methods were satisfactorily conducted and reported (although not compliant with the ORION guidelines; see Section 4.3); microbiological methods were less frequently scored as satisfactory and good (no use of neutralizers). Assessments of risk factors, reservoir, transmission, and root cause analysis scored from satisfactory to mostly good, apart from a few poor scores. Cross-sectional surveys were less clear and informative in their titles and abstracts. Poor scores were noted for description of products (terminology and active ingredients) and their indications for use. Methodological shortcomings included sample selection and denominators.

External validity: Generalizability could be affected by underreporting (see Section 4.1), as well as by representativeness: most studies were conducted in centers with a functional clinical microbiology laboratory, which were probably not representative of underserved, remote, and rural settings where, in addition, water supply is unreliable [35]. Further, although the present review was not designed or powered to assess low- versus middle-income countries, low-income countries represented only one-third of the articles, where they have serious challenges, e.g., only one-quarter (26%) of healthcare facilities in sub-Saharan Africa have a basic environmental cleaning service in place [1]. In that regard, the absence of articles from central Africa and Oceania most probably represented non-detection and underreporting. In light of the threat of antimicrobial resistance—which particularly affects LMICs [100]—the multidrug resistance of several bacteria in the current outbreak investigations was of particular concern. However, given the low number of cross-sectional surveys that performed state-of-the-art antibiotic susceptibility testing, antimicrobial resistance was probably underestimated.

## 5. Best Practices to Mitigate the Risk of Bacterial Contamination

Table 8 lists the aggregated best practices to mitigate the risk of contamination of AS, DI, and HH products along their life cycles in a healthcare facility.

Most best practices are issued by international guidelines and evidence-based WHO documents. They are universally applicable (also in HICs). Most are also technically feasible but may be demanding regarding financial and human resources: as an example, appropriate dilution and manual procedures for reprocessing require time, energy, clean water, and intensive staff training [5,101]. Other recommendations face serious challenges, such as disposable containers, cartridges, and pumps. The mounted pump and dosing systems of many currently marketed dispensers designed for single use are difficult or impossible to reach for mechanical disinfection. They are, therefore, susceptible to biofilm formation, causing subsequent contamination of freshly added soap [102]. Although it goes beyond the scope of this review, retrieving or defining the period-after-opening of many products may be difficult, as this may be branded-product-related. A useful overview of period-after-opening can be found in [103], although it should be noted that the data are defined for in-use conditions in HICs.

As in other aspects of IPC, the selection and prioritization of best practices is risk-based and relies on the leadership of IPC committees and the commitment of the management in healthcare facilities [5,104].

## 6. Outstanding Issues and Research Questions

Based on the overview analysis above, many outstanding issues and research questions about the contamination of AS, DI, and HH products in LMICs are similar to those from HICs [9], but many have different weights and relevance. As an example, underreporting

and its associated causes are relevant for HICs but are even more an issue in LMICs, as are investments to provide microbiologically safe water and biofilm control. Further, market access and financial affordability interfere with choices and procedures, as shown by the stock rupture of unit-dose alcohol pads implemented as a control measure for an outbreak of *Burkholderia cepacia* bloodstream infection [53]. Other examples of economic considerations in LMICs are, for instance, the preference of concentrated products (which subsequently need dilution) over ready-to-use products such as iodophors and favoring liquid soap products over alcohol-based handrubs [67,105].

Among outstanding issues specifically relevant for LMICs, field-adapted reusable containers and dispensers stand out given economic and ecologic concerns (waste management with incineration on site). Finding appropriate containers for alcohol-based handrub is a challenge [5,106], and inappropriate or malfunctioning dispensers are a barrier to hand hygiene roll-out and adherence [107,108]. Likewise, reprocessing is challenging (see Section 4.5), and reused dispensers suffered from pump or cap damage [107]. Proposed solutions were the selection of properly designed and affordable, good-quality dispensers and improved reprocessing of containers [107]. However, to the best of our knowledge, a good-quality affordable, durable, reusable container has not yet been marketed. Drafting a target product profile (as performed for use in HICs [109]) may fuel quality and low-cost mass production.

In-country production of AS, DI, and HH products can improve access. A recent experience of transfer of production of CHG gel (used in neonatal care [110]) to Kenya showed that even upscaled production of a difficult-to-manufacture product such as CHG gel [111] can be successfully achieved through global public–private partnerships (production technology transfer); the engagement of regulatory authorities, governments (List of Essential Medicines and therapeutic guidelines), and non-governmental organizations (advocacy, sensitization, and training); and exchanges with healthcare providers [112].

Compared to liquid soaps, bar soaps were more frequently associated with bacterial contamination in higher proportions than liquid soap products and sometimes with high bacterial counts. Similar findings are reported from HICs [9], and for this reason, liquid soap is preferred over bar soap in healthcare facilities [5]. However, transmissibility of bacteria during washing with bar soap was low when sufficient, good-quality water was used [113,114], and so far, outbreaks associated with bar soap have not been reported. Further, bacterial contamination of bar soap can be limited by keeping the bar soap dry [73]. In conclusion, a place for bar soap in lower-risk areas of healthcare facilities in LMICs may be considered.

Particularly in remote and rural areas, in-hospital preparation of products is applied. For in-hospital preparation of alcohol-based handrubs, detailed, user-friendly procedures were published [26]. For other products, such as producing 0.5% chlorine solutions, different methods can be used, such as the electrolysis of NaCl, dilution of bleach products, or dissolving of calcium hypochlorite tablets in water. Given the risks of miscalculations [79] and safety, generic guidelines and procedures are welcome.

Although not addressed in the articles from LMICs, exploring users' perceptions, attitudes, and beliefs about AS, DI, and HH products may be valuable to understand and influence practices. Articles from HICs noted that healthcare workers incorrectly perceived antiseptics as sterile [9], and in two articles from LMICs, similar referrals were made as to the expected sterility of the products [77,78]. Social sciences studies should also be conducted to understand and correct healthcare providers' deviations from procedures and to counter institutionalized practices that persist despite awareness and knowledge [94,115].

## 7. Limitations and Strengths

Among the limitations discussed above, the greatest impact possibly arose from underreporting. This was linked to the limited availability of clinical microbiology laboratories, poor awareness, underdetection, and lack of time or expertise. Further, broad research questions and heterogeneity of the results precluded a meta-analysis. Finally, best practices

were compiled from articles and guidelines but were not scored for evidence or expressed according to Grading of Recommendations Assessment, Development and Evaluation (GRADE) methodology [116]. We believe that GRADE scoring and recommendations are a next step toward the implementation of risk-based mitigation of contamination of AS, DI, and HH products. As for the strengths, they were the in-depth review with the specific analysis of microbiology techniques and the inclusion of hand hygiene products (particularly liquid soap) in the search for cross-sectional surveys.

## 8. Conclusions

Although underreported, contaminated AS, DI, and HH products may act as reservoirs of healthcare-associated outbreaks in LMICs, with substantial case–fatality ratios and associations with specific risk factors. Domains of implementation research include field-adapted containers and reprocessing, in-country production, water quality, biofilm control, healthcare provider practices, and the role of bar soap.

**Table 6.** Overview of risk of bias and quality of methods of (pseudo)outbreak investigations demonstrating bacterial contamination of antiseptics, disinfectants, and hand hygiene products in healthcare facilities in low- and middle-income countries. The colors refer to the categories of "good", "satisfactory", and "poor", as described in Supplementary Table S1. Grey color means "not conducted" or "not applicable". The sign "*" is added in colored circles when a neutralizer was used.

| Item Scored | Thong 1981 [55] | Anyiwo 1982 [47] | Cissé 1987 [49] | Kaitwatcharachai 2000 [75] | Parasakthi 2000 [51] | Nasser 2004 [53] | Espinosa de los Monteros 2008 [54] | Ben Saida 2009 [77] | Khanna 2013 [76] | Stoesser 2015 [48] | Valderrama-Beltrán 2019 [44] | Clara 2021 [52] | Said 2022 [56] |
|---|---|---|---|---|---|---|---|---|---|---|---|---|---|
| The healthcare setting was well-described | green | red | green | green | green | green | green | yellow | green | green | green | green | green |
| Title and Abstract provided information about the "outbreak" or "cross-sectional survey" | green | red | green | green | green | green | green | green | green | green | green | green | green |
| The outbreak setting was well-described | yellow | red | yellow | yellow | yellow | yellow | yellow | yellow | yellow | yellow | yellow | green | yellow |
| The (pseudo)outbreak was well-described | green | red | green | green | green | green | yellow | green | yellow | green | green | green | green |
| Outbreak index organism microbiological methods were described in sufficient detail | green | yellow | yellow | green | green | green | green | yellow | green | green | red | green | green |
| The outbreak investigation included a clinical epidemic investigation | green | yellow | yellow | green | yellow | green | yellow | green | red | red | green | green | yellow |
| The outbreak environmental investigation was oriented by the clinical epidemic investigation | green | yellow | green | green | green | green | green | green | red | red | green | green | green |
| Product and active ingredients were provided | green | green | red | green | green | green | green | green | green | green | green | green | green |
| The use and application of the product(s) were well-described | green | green | green | green | green | green | red | green | green | green | green | green | green |
| Correct terminology was used (antiseptics, disinfectants, hand hygiene products) | green | green | green | green | green | green | red | green | green | green | green | green | green |

**Table 6.** *Cont.*

| Item Scored | Thong 1981 [55] | Anyiwo 1982 [47] | Cissé 1987 [49] | Kaitwatcharachai 2000 [75] | Parasakthi 2000 [51] | Nasser 2004 [53] | Espinosa de los Monteros 2008 [54] | Ben Saida 2009 [77] | Khanna 2013 [76] | Stoesser 2015 [48] | Valderrama-Beltrán 2019 [44] | Clara 2021 [52] | Said 2022 [56] |
|---|---|---|---|---|---|---|---|---|---|---|---|---|---|
| Microbiological culture methods used were appropriate and well-described | 🟢 | 🟡 | 🔴 | 🟢* | 🟡 | 🟢 | 🔴 | 🟢 | 🔴 | 🔴 | 🔴 | 🟢 | 🟡 |
| Antibiotic susceptibility testing methods were reported and appropriate | ⚪ | ⚪ | 🟡 | 🟡 | 🟢 | 🟢 | 🟢 | 🟢 | 🟢 | 🟢 | 🟢 | ⚪ | 🟢 |
| Microbiological typing methods used were appropriate | 🟡 | 🔴 | 🟡 | 🟢 | 🟢 | 🟢 | 🟢 | 🟢 | 🔴 | 🟢 | 🟢 | 🟢 | 🟡 |
| Reporting of results was complete and appropriate | 🟢 | 🟢 | 🟢 | 🟢 | 🟢 | 🟢 | 🟢 | 🟢 | 🟢 | 🟢 | 🟢 | 🟢 | 🟢 |
| Risk factors were assessed | 🟢 | 🟢 | 🟢 | 🟢 | 🟢 | 🟢 | 🟡 | 🟢 | 🟢 | 🔴 | 🟢 | 🟢 | 🟢 |
| Additional investigations for risk factors were conducted and reported (interview, questionnaire, review of procedures) | 🟢 | 🟡 | 🟢 | 🟢 | 🟢 | 🟢 | ⚪ | ⚪ | ⚪ | ⚪ | 🟢 | 🟢 | 🟢 |
| Evidence for reservoir was assessed | 🟢 | 🟡 | 🟢 | 🟢 | 🟢 | 🟢 | 🟢 | 🟢 | 🟡 | 🟢 | 🟢 | 🟢 | 🟢 |
| Evidence for transmission was assessed | 🟡 | 🟡 | 🟢 | 🟢 | 🟢 | 🟢 | 🔴 | 🟢 | 🟢 | 🟢 | 🟢 | 🟢 | 🟢 |
| Evidence for root cause was assessed | 🟢 | 🟢 | 🟡 | 🟢 | 🟢 | 🟢 | 🔴 | 🟢 | 🟢 | 🔴 | 🟢 | 🟢 | 🟢 |

**Table 7.** Overview of risk of bias and quality of methods of cross-sectional surveys demonstrating bacterial contamination of antiseptics, disinfectants, and hand hygiene products in healthcare facilities in low- and middle-income countries. The colors refer to the categories of "good", "satisfactory", and "poor", as described in Supplementary Table S1. Grey color means "not conducted" or "not applicable". The sign "*" is added in colored circles when a neutralizer was used.

Legend: G = green ("good"), Y = yellow ("satisfactory"), R = red ("poor"), grey = "not conducted"/"not applicable", * = neutralizer used.

| Item Scored | Khor 1977 [45] | Olayemi 1994 [78] | Danchaivijitr 1995 [82] | Kajanahareutai 1995 [79] | Keah 1995 [70] | Arjunwadkar 2001 [81] | Ogunsola 2002 [57] | Gajadhar 2003 [60] | Danchaivijitr 2005 [58] | Subbannayya 2005 [65] | Tytler 2006 [59] | Afolabi 2007 [73] | Zhang 2008 [68] | El-Mahmood 2009 [61] | Muchina 2009 [69] | Zeiny 2009 [66] | Aktas 2010 [62] | Caetano 2011 [71] | Deress 2014 [74] | Singh 2014 [80] | Akabueze 2015 [63] | Biswal 2015 [67] | Salma 2016 [84] | Altaher 2016 [72] | Firesbhat 2021 [64] |
|---|---|---|---|---|---|---|---|---|---|---|---|---|---|---|---|---|---|---|---|---|---|---|---|---|---|
| Healthcare setting was well-described | Y | Y | Y | Y | Y | Y | G | Y | Y | Y | Y | G | Y | G | G | G | G | G | G | Y | Y | Y | Y | G | G |
| Title and Abstract provided information about the "outbreak" or "cross-sectional survey" | R | R | R | R | R | R | R | G | R | R | G | R | R | R | G | R | G | G | R | R | R | R | R | R | R |
| Objectives were clearly described | G | G | G | R | G | G | G | G | G | G | G | G | G | R | G | G | G | G | G | G | G | G | G | G | G |
| Product and active ingredient were provided | R | G | Y | G | Y | G | G | G | G | G | R | G | G | Y | G | G | G | Y | G | G | G | G | G | G | G |
| The use and application of the product(s) were well-described | G | R | R | R | R | R | G | R | G | R | G | R | R | R | G | G | G | R | G | G | G | G | G | G | G |
| Correct terminology was used (antiseptics, disinfectants, hand hygiene products) | G | G | G | G | G | G | Y | G | R | R | G | G | G | G | G | G | G | R | G | G | G | G | G | G | G |
| Sample selection and numbers (denominators) were provided | R | G | G | Y | G | G | Y | G | G | G | R | R | R | R | G | G | G | G | G | Y | G | R | R | R | G |
| Sufficiently large sample sizes were provided | Y | R | G | G | Y | G | G | G | G | G | Y | R | R | R | G | G | G | G | G | G | G | G | G | G | G |
| Microbiological culture methods used were appropriate | G* | Y | R | G | G | Y | G* | G* | Y | G* | Y* | Y | Y* | Y* | Y | Y | Y | G* | Y | G | G | Y | Y | Y | Y |
| Antibiotic susceptibility testing methods were reported as appropriate | grey | grey | grey | grey | grey | grey | grey | grey | grey | grey | grey | Y | G | R | grey | grey | grey | grey | Y | grey | grey | grey | grey | grey | G |
| Additional investigations were conducted and well-reported (interview, questionnaire, review of procedures) | R | R | G | R | Y | Y | G | G | G | G | G | G | G | G | G | G | G | G | G | G | Y | Y | G | G | R |
| Reporting of results was complete and appropriate | G | Y | G | G | G | G | G | G | G | G | G | Y | G | G | G | G | G | G | G | G | G | G | G | G | G |

**Table 8.** Best Practices to mitigate the risk of bacterial contamination of antiseptics, disinfectants, and hand hygiene products used in healthcare facilities in low- and middle-income countries. Note that the table does not list recommendations about choice, characteristics (e.g., expiry date or date-after-opening), and use of individual products. Abbreviations: IPC = Infection Prevention and Control; WHO = World Health Organization.

| Best Practices/Recommendations | References |
|---|---|
| **General recommendations** | |
| ○ Antiseptics, disinfectants, and hand hygiene products are part of national and health facilities' IPC programs (Core Components 1, 6, and 8 of the WHO Core Components of Infection Prevention and Control) <br> ○ Assure a Quality Management System along the lifecycle of products in healthcare facilities overseen by the IPC Committee (procurement, inventory and stock, personnel, procedures, equipment, processes, documents and records, assessment, and monitoring) <br> ○ Implement according to the WHO multimodal strategy (Core Component 5) <br> ○ Follow the instructions of the manufacturer for preparation, concentration, safety, use, etc. <br> ○ Adhere to a risk-based approach with actions and means proportional to patient risk groups (neonatology, oncology, etc.) and risk procedures | [60,117–119] |
| **Selection, procurement, reception control, stock management, and supplier evaluation** | |
| ○ Limit numbers of different products; harmonize and standardize used products <br> ○ Have a final responsible person (e.g., pharmacist) <br> ○ Choose well-known and qualified product, brand, and distributor/supplier <br> ○ Perform reception and preproduction control: concentration and microbiological quality <br> ○ Record issues with products and organize supplier evaluation <br> ○ Define period-after-opening | [5,60,71,102,103,109, 117,120,121] |
| **Container and dispenser characteristics** | |
| ○ Preferably, use original and disposable containers; limit use of reprocessed containers <br> ○ Design and functioning prevent contamination during in-use and allow reprocessing <br>    ● Containers with dispensing systems <br>      ✓ Disposable dispenser (single-use pump) preferred <br>      ✓ Must be accessible for cleaning and disinfection <br>      ✓ Sealed disposable refill cartridges <br>      ✓ Cartridges with integrated nozzle (gravitational dispenser) <br> ○ Lid and stopper must be present and fit leakage-free <br>    ● Must be present and fit tightly <br>    ● Avoid cork stoppers of liners <br> ○ Hand-free-actioning dispenser is preferable over hand-commanded dispenser; elbow-command is preferred over sensor-operated command | [109,122,123] |

**Table 8.** *Cont.*

| Best Practices/Recommendations | References |
|---|---|
| ○ If reprocessing and reuse are considered (economic or ecologic reasons)<br>    ● Preference for reusable containers (designed for reuse and withstanding reprocessing)<br>    ● Avoid recycling of containers designed for other purposes (e.g., soft drink bottles)<br>○ Use small- to medium-volume containers for in-use products to assure short in-use durations Examples of recommended volumes<br>    ● 300–500 mL for table-top dispenser with well-functioning pump<br>    ● 100 mL for an individual pocket dispenser<br>○ Visibility of label and content to allow easy identification of the product<br>○ Outside surface easy to clean and disinfect | [109,122,123] |

**Preparation process (dilution, bottling, and labeling)**

| | |
|---|---|
| ○ Work in an assigned and clean preparation room; before starting preparation, clean working surfaces with a detergent, and disinfect benches with 0.5% chlorine<br>○ Assure trained and competent staff (no students or aids), job description, and supervision<br>○ Rigorously wash all materials needed for preparation and dilution: bottles, measuring cylinders and jugs, plastic or metal funnels (if metal funnel, stainless steel is better), final containers, and alcohol meters<br>○ Rinse materials with clean tap water or preboiled water, and use freshly prepared sterile distilled water if available; note: filtering water is not always effective for sterilization<br>○ For making dilutions, use clean water, e.g., freshly boiled water, cooled water to room temperature, or if available, use freshly prepared sterile distilled water<br>○ After preparation, pour the solution in small containers (previously washed and dried; see above) without touching the rim or the solution itself<br>○ Label each container (product name and concentration, preparation date, lot number, expiry date (after dilution), shelf life, initials of operator); provide a place to write "day of in-use" and period-after-opening or expiry date after opening<br>○ For dilutions (e.g., water-based antiseptics or disinfectants such as chlorine and QUAT), prepare working solutions immediately before use and discard any remaining solution | [5,53,60,61,70,84,103, 115,124–128] |

**Storage and Distribution**

| | |
|---|---|
| ○ Limit stock in wards<br>○ Keep products away from direct sunlight, ignition, and hot sources (particularly in cases of alcohol)<br>○ Keep storage rooms well-ventilated and protected from dust and high or low temperatures | [5,70,103] |

**In-use**

| | |
|---|---|
| ○ At first use, write opening date and period-after-opening or expiry date after opening date<br>○ Timely remove and replace expired products<br>○ Recap containers after taking the needed amount of disinfection or antisepsis | [5,27,65,66,71,73,103, 109,128–134] |

**Table 8.** *Cont.*

| Best Practices/Recommendations | References |
|---|---|
| ○ Prevent retrograde contamination of containers while in-use<br>    ● Keep containers closed (tight-fitting stopper)<br>    ● Keep hand rub container dispensers away from palm when pouring for hand hygiene<br>    ● Ensure at the handwash station enough space between the dispenser and the sink (hands should freely move between dispenser, tap outlet, and sink without touching them<br>    ● Do not touch dispenser nozzle or bottleneck with hands or cotton or gauze balls<br>    ● When you need to dip cotton or gauze in an antiseptic or disinfectant, pour an aliquot in a small-volume container for immediate use; do not dip the cotton or gauze in the container<br>    ● Do not soak cotton balls or gauze in antiseptics and disinfectants<br>    ● Do not top-up (top-off or refill) a container<br>○ Clean and disinfect the external surface of the container (which is a high-touch surface)<br>○ Protect bar soap used for hand hygiene from humidity<br>    ● Use a porous rack, a perforated receptacle, or a receptacle with a second layer enabling water drainage<br>    ● Use of individual or small pieces of bar soap | [5,27,65,66,71,73,103,<br>109,128–134] |
| **Container reprocessing** | |
| ○ Assign an area for reprocessing; use trained and competent staff<br>○ Thermal reprocessing with professional washer disinfector is mostly not available → manual processing<br>○ Manual processing<br>    ● Empty the reusable container and rinse abundantly first to remove product remnants<br>    ● Clean with a non-antimicrobial detergent, use a brush for areas difficult to reach, and rinse with clean water (freshly prepared sterile distilled water (preference) or preboiled and cold tap water)<br>    ● Sterilize or disinfect the container: autoclaving has preference; if not possible, chemical disinfection<br>    ● Autoclaving: 15 min, 121 °C<br>    ● Chemical disinfection<br>        ✓ Soak in 0.5% chlorine for 15 min<br>        ✓ Rinse with clean water (see above)<br>        ✓ Let the container dry and store it closed in a dust-free environment until reuse<br>○ Note: container materials that are autoclavable and not autoclavable<br>    ● Autoclavable: polypropylene (PP), polypropylene copolymer (PPCO), fluoropolymer (Teflon PFA, FEP, or ETFE) and polycarbonate (PC)<br>    ● Non-autoclavable: high-density polyethylene (HDPE), low-density polyethylene (LDPE), polyethylene terephthalate (PET), polyethylene terephthalate co-polyester (PETG) | [5,71,101,107–<br>109,128,130,132] |

**Supplementary Materials:** The following supporting information can be downloaded at: https://www.mdpi.com/article/10.3390/hygiene3020010/s1, Table S1: Method used to appraise risk of bias and quality of methods and reporting of (pseudo)outbreaks and cross-sectional surveys related to contaminated antiseptics, disinfectants, and hand hygiene products used in healthcare facilities in low- and middle-income countries; Table S2: Overview of healthcare-associated (pseudo)outbreaks in low- and middle-income countries associated with bacterial contamination of antiseptics, disinfectants, and hand hygiene products; Table S3: Overview of cross-sectional studies assessing bacterial contamination of antiseptics and disinfectants used in healthcare facilities in low- and middle-income countries; Table S4: Overview of cross-sectional surveys assessing bacterial contamination of soaps used for hand hygiene in healthcare settings in low- and middle-income countries; Table S5: Other results: Articles demonstrating bacterial contamination of antiseptics, disinfectants, and hand hygiene products with another fomite identified as the actual reservoir and source of infection; Document S1: PROSPERO Protocol; Document S2: Data extraction.

**Author Contributions:** Conceptualization, P.L. and J.J.; methodology, P.L., B.V.d.P., A.-S.H. and J.J.; validation, P.L., A.-S.H. and J.J.; database screening and data extraction, P.L., B.V.d.P. and J.J.; data analysis, P.L.; writing—original draft preparation, P.L. and J.J.; writing—review and editing, P.L., A.-S.H., E.A., B.V.d.P., V.K., C.M.G.K., A.Z., H.T., D.A. and J.J.; supervision, J.J. All authors have read and agreed to the published version of the manuscript.

**Funding:** This research was funded by the Belgian Directorate for Development Cooperation and Humanitarian Aid (DGD) and the Institute of Tropical Medicine, Antwerp, Belgium. PL received a PhD fellowship from DGD.

**Institutional Review Board Statement:** Not applicable.

**Informed Consent Statement:** Not applicable.

**Data Availability Statement:** Data extracted from the original articles supporting our results can be found in Supplementary Document S2 submitted with this work.

**Acknowledgments:** We would like to thank our colleagues from the Institute of Tropical Medicine, Antwerp, for their contribution to the literature search. We particularly thank Nic Peeters from the institute's library, as well as Peter Hyland and Marjan Peeters from the Tropical Bacteriology Unit.

**Conflicts of Interest:** The authors declare no conflict of interest.

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
