# Peer review of "Bacterial Contamination of Antiseptics, Disinfectants, and Hand Hygiene Products Used in Healthcare Settings in Low- and Middle-Income Countries—A Systematic Review"

_2673-947X, doi:10.3390/hygiene3020010_

Round 1
Reviewer 1 Report
Thank you for this invitation to review the manuscript entitled “Bacterial contamination of antiseptics, disinfectants and hand 2 hygiene products used in healthcare settings in low- and mid- 3 dle-income countries – a systematic review”
The work is very good and the efforts are obvious, however, I have a few comments hope you could consider while you are revising your manuscript.
For comparison between HIC and LMIC countries, can you test for significance to find if the differences and similarities are significant or not? (e.g. lines 213:225; page 5)
page 5, lines 216: “The outbreaks’ median (range) duration was 216 12 (1 – 365) weeks”. Are you sure about this range?
page 5, lines 226: 228: “Ten articles reported patients’ outcome: median (range) case fatality ratio was 26.0% 226 (0.0 - 88.5%); this figure was higher than observed in HIC (median 9.2% (00.0 - 60.0%)). 227 Median (range) number of deaths was 4 (1 to 8).” Please revise the ranges, the minimum can’t be 1 when represented in numbers and 0 when represented as percentage.
page 10, lines – ------- “Concentrations of products assessed in outbreak articles varied: for water-based CHG, they ranged from a 1/2000 dilution [44] to 2% and 4% respectively [41,49]” to which you refer by the values “2% and 4%”?
page 11, lines 1 “the differences were 20-fold and nearly 4-fold,” can you rewrite for more clarification?
Table 2B footnote: replace “water CHG” by “water-based CHG”
page 14, lines 157: “wide-type”, do you mean “wild-type”?
Tables 3A and 3B, if possible to add references to the tables
page 20, lines – ------- “All but 1 surveys”. Numbers from 1-9 should be written in letters
Table 4A & Table 4B. you should describe the full color code used in the table; red, green, yellow and grey in the table footnote
The sign “+” is very very small needs eagle’s eye to find it.
In the same tables you should provide the reference number as mentioned in the reference list
Table 5. please provide notes and abbreviations at the table footnotes
Author Response
Reply to comments of Reviewer 1:
Point 1: The work is very good and the efforts are obvious, however, I have a few comments hope you could consider while you are revising your manuscript.
Response 1: We thank the reviewer for this encouraging comment.
Point 2: For comparison between HIC and LMIC countries, can you test for significance to find if the differences and similarities are significant or not? (e.g. lines 213:225; page 5)
Response 2: We thank the reviewer for this comment and calculated and added the statistical significance of the items which was visibly different between results from high-income countries versus low-and middle-income countries respectively. We asked expert statistical advice which recommended to make the comparison by aggregated case-fatality ratio. The revised sentence (Revised Version with Track Changes is as “Ten articles reported patients’ outcome: median (range) case fatality ratio was 26.0% (0.0 - 88.5%), aggregated case fatality ratio was 9.1%; this figure was significantly higher than observed in HIC (median 0.0% (0.0 – 60.0%), aggregated 6.1%, p = 0.027, chi square).” Page 5, Lines 244-246. The other outbreak characteristics (duration, numbers of affected patients, infected body sites and specimens cultured) were comparable between HIC and LMIC, reason why we did not test for statistical significance.
Point 3: page 5, lines 216: “The outbreaks’ median (range) duration was 12 (1 – 365) weeks”. Are you sure about this range?
Response 3: We thank the reviewer for his/her comment. The outbreak duration was converted in weeks, and the median was indeed 12 ranging between 1 week to 365 weeks; The extreme duration (365 week) was observed in an outbreak report of Burkholderia cepacia bloodstream infections in Lebanon extended from February 1993 to May 2000. To avoid confusion for the reader, we replaced “365 weeks” by 7.25 years: See Page 5 Lines 231 of the Revised Version Track Changes: “The outbreaks’ median (range) duration was 12 weeks (1 week – 7.25 years), …”
Point 4: page 5, lines 226: 228: “Ten articles reported patients’ outcome: median (range) case fatality ratio was 26.0% 226 (0.0 - 88.5%); this figure was higher than observed in HIC (median 9.2% (00.0 - 60.0%)). 227 Median (range) number of deaths was 4 (1 to 8).” Please revise the ranges, the minimum can’t be 1 when represented in numbers and 0 when represented as percentage.
Response 4: We thank the reviewer for this relevant comment. The apparent contradiction is caused by the fact that the median range of deaths was calculated for the 9 out of 10 outbreak articles which provided data about case fatalities. To correct for this confusing phrasing were rephrased the paragraph as follows (Page 5, Lines 240 - 243 of the Revised Version Track Changes): “Ten articles reported patients’ outcome: median (range) case fatality ratio was 26.0% (0.0 - 88.5%); this figure was higher than observed in HIC (median 0.0% (0.0 - 60.0%)). In one outbreak, no case fatality was recorded, median (range) number of deaths for the remaining 9 articles which reported outcome was 3 (1 - 23). In 3 outbreaks, case-fatality ratios were ≥ 40% [49,51].”
Point 5: page 10, lines 326 -327 ------- “Concentrations of products assessed in outbreak articles varied: for water-based CHG, they ranged from a 1/2000 dilution [44] to 2% and 4% respectively [41,49]” to which you refer by the values “2% and 4%”?
Response 5: We thank the reviewer for his question and comment. In the included articles, the product concentrations were expressed diversly. Certain papers used dilutions (and other used concentrations; 2% and 4% working CHG concentrations which are the two highest product concentrations of manufactured and marketed products. To clarify this, we revised the text as follows (Page 10, Lines 327-334 of the Revised Version Track Changes): “Concentrations of products assessed in outbreak articles were expressed either as dilutions or concentrations and varied: for water-based CHG, they ranged from a 1/2000 dilution of a 5% stock solution [45] to 2% and 4% respectively (which are among the highest concentrations of products marketed) [42, 50]. In one article, a 1/2000 dilution of a 1.5% CHG-QUAT solution was contaminated [74]”.
Point 6: page 11, lines 1 “the differences were 20-fold and nearly 4-fold,” can you for more clarification?
Response 6: We thank the reviewer for his/her suggestion. To make the sentence more comprehensible we have rewritten it as follows (Page 11, Lines 365-367 of the Revised Version Track Changes): “… in 2 of them (assessing > 40 samples for each bar and liquid samples), the proportions of contaminated bar soap were 4-fold and 20-fold higher than in liquid soap (30/50 versus 7/44 and 61/99 versus 2/60 respectively) (Supplementary Table 4) [64,65].”
Point 7: Table 2B footnote: replace “water CHG” by “water-based CHG”
Response 7: We thank the reviewer for this comment and made the correction ((Page 11, Table 2B - footnote of the Revised Version Track Changes).
Point 8: page 14, lines 157: “wide-type”, do you mean “wild-type”?
Response 8: Indeed, we thank the reviewer and made the required correction (Page 14 Lines 551 of the Revised Version Track Changes).
Point 9: Tables 3A and 3B, if possible, to add references to the tables
Response 9: We thank the reviewer for his/he suggestion. However, adding references to Tables 3A and 3B is not feasible because in some studies many bacterial species were isolated. We provided the details regarding bacterial species reported in each study in the Supplementary Tables 2, 3 and 4. We added to the Legend of the Tables 3A and 3B (Page 15, Line 560-561 and Page 16, Line 572 of the Revised Version Track Changes): “Details about the isolates (bacterial load, references) can be found in Supplementary Tables 2” and “Details about the isolates (bacterial load, references) can be found in Supplementary Table 3 and 4”, respectively.
Point 10: page 20, lines – ------- “All but 1 surveys”. Numbers from 1-9 should be written in letters
Response 10: We thank the reviewer for his/her suggestion. The Hygiene author instructions do not mention to write numbers from 1 to 9 in letters, so we used the Arabic numbering across the manuscript. We however agree with the reviewer that in this case, the sentence reeds better when using “one” so we replaced “1” by “one” (Page 10, Lines 317, Page 12, Line 399, Page 13, line 453 and Page 20, Line 688 of the Revised Version Track Changes).
Point 11: Table 4A & Table 4B. you should describe the full color code used in the table; red, green, yellow and grey in the table footnote
The sign “+” is very very small needs eagle’s eye to find it.
In the same tables you should provide the reference number as mentioned in the reference list
Response 11: We thank the reviewer for these relevant comments. As it is essential information to understand the Table 4A and 4B, we preferred to add this explanation in the title’s legend rather than in the footnote. In addition, we gave the details of the scoring procedure in the Supplementary Table 1 to help the reader to better understand the bias assessment. We added the reference numbers for the column representing the articles and changed the “+” sign to a more visible symbol “*”.
Point 12: Table 5. please provide notes and abbreviations at the table footnotes
Response 12: We thank the reviewer for his/her comment. We would like to specify that the last line in the Table 5 “Note” is not intended as footnote of the table but refers to the cell and topic about the containers’ description, and we assumed that it is better to keep it as part of the table. To make it clearer, we slightly adapted the text as follows: “Note: container materials which are autoclavable and not autoclavable are:” (Page 28 Table 5 of the Revised Version Track Changes)

Reviewer 2 Report
Comments to the Author
It is a well-written and important paper.
General remark:
The study covers a relevant topic and is well-written in most parts. However, the manuscript has some major limitations. This systematic review and meta-analysis do not fulfill the requirements of the Preferred Reporting Items for Systematic Reviews and Meta-Analyses guidelines (see below).
- Methods: The authors state that the review follows the Preferred Reporting Items for Systematic Reviews and Meta-Analyses (PRISMA) guidelines. However, the manuscripts lack many items from the checklist.
I clearly suggest that the authors check their manuscript for all items of the PRISMA checklist and revise the manuscript accordingly.
Please define the inclusion and exclusion criteria.
Author Response
Reply to comments of Reviewer 2:
Point 1: The study covers a relevant topic and is well-written in most parts. However, the manuscript has some major limitations. This systematic review and meta-analysis do not fulfil the requirements of the Preferred Reporting Items for Systematic Reviews and Meta-Analyses guidelines (see below).
- Methods: The authors state that the review follows the Preferred Reporting Items for Systematic Reviews and Meta-Analyses (PRISMA) guidelines. However, the manuscripts lack many items from the checklist.
- I clearly suggest that the authors check their manuscript for all items of the PRISMA checklist and revise the manuscript accordingly.
Response 1: We thank the reviewer for this suggestion. We have checked the compliance of our manuscript with the Preferred Reporting Items for Systematic Reviews and Meta-Analyses (PRISMA) guidelines (PRISMA2020 Checklist (http://prisma-statement.org/documents/PRISMA_2020_checklist.pdf?AspxAutoDetectCookieSupport=1), PRISMA2020 Abstracts Checklist (http://www.prisma-statement.org/documents/PRISMA_2020_abstract_checklist.pdf).
The PRISMA checklist is primarily designed for interventional studies and meta-analysis of specific research questions. The objectives of the present review were multiple, as were the format and information provided by the data sources (45 years overview, mix of articles types (letters, short communications, original research)). Despite this, the manuscript meets the following PRISMA criteria: 1 (Title), 3 (Rationale), 4 (Objectives), 5 Eligibility criteria (both 4 and 5 have been better formulated upon the reviewer’s suggestion, see below), 6 (Information Sources), 7 (Search strategy), 8 (Selection process), 9 (Data collection process), 10 (Data items), 11 (Study risk of bias assessment), 13 (Synthesis methods) and 14 (Reporting bias assessment). As to the PRISMA 2020 Abstracts checklist (Criterium 2), the present Abstract complies with all criteria except for citing the funding source (reasons of word count).
Given the multitude of data, a meta-analysis was not possible (mentioned in the Discussion Page 30 Lines 824 - 825) “Further, broad research questions and the heterogeneity of the results precluded a meta-analysis”. Neither it was possible to express risk of bias in a quantitative way, and for this reason we constructed a qualitative checklist based on the ORON and MICRO checklist, which was acknowledged as appropriate in the Reviewer’s comment 6.
In conclusion, upon the reviewer’s comment, we checked the manuscript for the criteria of the PRISMA checklist, adapted the text for criteria Objectives and Eligibility Criteria (4 and 5 respectively) and we can confirm the other criteria are met; we further justified the absence of a meta-analysis.
Point 2: Please define the inclusion and exclusion criteria.
Response 2: We thank the reviewer for this comment. We adapted the text as follows (Revised Version Track Changes Page 3, lines 62 - 68): “Inclusion criteria were original research studies addressing bacterial contamination of AS DI and HH products, comprising (pseudo-)outbreak investigations and cross-sectional surveys conducted in healthcare facilities in LMIC. Exclusion criteria were: editorials, reviews, studies limited to molecular typing, experimental studies, studies in the community and veterinary healthcare, studies addressing exclusively bacterial contamination of water, sinks or other sanitary equipment and studies from high-income countries.”

Reviewer 3 Report
1. A brief summary
I appreciate the opportunity to review this interesting manuscript. The article entitled “Bacterial contamination of antiseptics, disinfectants and hand hygiene products used in healthcare settings in low- and middle-income countries – a systematic review” is eligible for publishing consideration in the Hygiene. This investigation may confirm that contaminated antiseptics, disinfectants and hand hygiene productsact as a reservoir for outbreaks linked to healthcare that have a substantial case fatality, and the best practices can lower the possibility of contamination. However, there are major concern about this manuscript.
2. General concept comments
Comment 1. The manuscript is moderately structured and written. It has a crucial clinical message and ought to pique the readers' interest. Unfortunately, the citing recent references published within the last 5 years (2018-2023) is just 33.62% (38/113 publications). Please consider updating the references.
Comment 2. Please double-check the citation formatting for each item listed at the end of the manuscript.
Comment 3. Some keywords (“intrinsic” and “in-use”) may not be necessary. Since the unspecific keywords do not adequately reflect its contents.
Comment 4. Please arrange the keywords alphabetically for a standardized presentation.
Comment 5. Your introduction section needs more detail. I suggest that you improve the statement of clinical relevance.What is the significance of this review for clinical practice, and what is new in this study?
Comment 6. The included studies' quality assessments were performed appropriately, and the results were reported in a manner that was consistent with the procedures used to extract the data that were discussed in the material and methods section. Unfortunately, the Best practices were not assessedfor certainty of the evidence using Grading of Recommendations Assessment, Development and Evaluation (GRADE)
Comment 7. In tables 2B and 3B, the term "H2O2" should be used instead of "hydrogen peroxide, HP" to be consistent with all of the other generic names.
Comment 8. The subsectionof 4.8 (Internal and external validity)have a different format, the subheadingof the sectionis written in boldletters at the beginning.
Comment 9. The author's clarity in stating "(6) Outstanding issues and research topics and (7) Limits and Strengths" is remarkable. However, the conclusion section of this manuscript is quite vague and lengthy. Some conclusions should state the primary point of the obtained major content and make recommendations for clinical practice. Additionally, this research's conclusion section requires a suggestion for further research.
Comment 10. Finally, I am grateful for the chance to review this submission. Although the topic is valuable and interesting, further interpretation and motivation is required. Wishing the authors success for their paper publication, I hope the suggestions are beneficial.

Author Response
Reply to comments of Reviewer 3:
Point 1: Comment 1. The manuscript is moderately structured and written. It has a crucial clinical message and ought to pique the readers' interest. Unfortunately, the citing recent references published within the last 5 years (2018-2023) is just 33.62% (38/113 publications). Please consider updating the references.
Response 1: We thank the reviewer for his/her comment. The reference list is indeed skewed by the spectrum of the articles retrieved according to the research questions but we are confident that the selection of other references (e.g. used in the Best Practices and Discussion section) is relevant and updated: in an extended search including several databases and grey literature, we found 38 source articles of which 4 were published since 2018, which is proportional to the entire 45-year time period of publications retrieved, i.e. from 1977 to 2022. Subtracting these articles from the overall 38/113 ratio mentioned by the reviewer gives a ratio of (38 – 4/113 – 34) = 34/79 or 48.1% referrals to publications since 2018. These publications include a review about antiseptics/disinfectants in Ebola virus settings (reference 6), recent documents about Infection Prevention & Control and Water, Sanitation & Hygiene published by WHO (references 1, 4, 10 and 23), lead publications about clinical bacteriology and antimicrobial resistance in LMIC (refs 9, 47, 94 and 97) and methodological papers (references 12, 24, 25, 29, 31, 92, 113).
Point 2: Comment 2. Please double-check the citation formatting for each item listed at the end of the manuscript.
Response 2: We thank the reviewer for this comment, we have checked and adapted the format of citation according to the format required by the journal.
Point 3: Comment 3. Some keywords (“intrinsic” and “in-use”) may not be necessary. Since the unspecific keywords do not adequately reflect its contents.
Comment 4. Please arrange the keywords alphabetically for a standardized presentation.
Response 3: We thank the reviewer for the relevant suggestions in comments 3 and 4, we have removed “intrinsic” and “in-use” from the keywords and ranked them alphabetically. The resulting list of key-words is as follows: “antiseptics; bacterial contamination; Best Practices; cross-sectional; disinfectants; hand hygiene; low- and middle-income countries; outbreak.” (Revised Version Track Changes Page 1, Lines 32-33).
Point 4: Comment 5. Your introduction section needs more detail. I suggest that you improve the statement of clinical relevance. What is the significance of this review for clinical practice, and what is new in this study?
Response 4: We thank the reviewer for this comment, we have revised the Introduction to detail the clinical relevance of the bacterial contamination of antiseptics, disinfectants and hand hygiene products based on the findings previously found in high-income countries. In addition, we highlighted what is new in our review compared to previous literature. The revised text reads as follows (Revised Version Track Changes Page 2, Lines 62 - 68): “The present systematic review differs from the previous narrative reviews (dating from before 2007) by its extensive search including grey literature and its appraisal of reporting quality and of risk of bias. Further, it includes soap products and assesses impact of contamination (numbers of patients affected and case-fatality ratios). It appraises the microbiological spectrum using updated nomenclature, the associated risk factors as well as attribution and trans-mission. Finally, Best Practices to mitigate contamination of AS DI and HH products are listed.”
Point 5: Comment 6. The included studies' quality assessments were performed appropriately, and the results were reported in a manner that was consistent with the procedures used to extract the data that were discussed in the material and methods section. Unfortunately, the Best practices were not assessed for certainty of the evidence using Grading of Recommendations Assessment, Development and Evaluation (GRADE)
Response 5: We thank the reviewer for this comment. Indeed, we have considered the options to appraise the Best Practices according to the GRADE system (https://www.gradeworkinggroup.org/). A GRADE process will put findings into practice taking into account scientific evidence and available means and priorities which may differ according to place, time and setting. But it requires appraisal and consensus by a multidisciplinary team of experts and is especially challenging in Infection Prevention Control as experimental studies and outcome measurements are not feasible. Despite its relevance, we therefore believe that GRADE recommendations are outside the scope of this review and are a next and separate step into implementation. We have acknowledged this limitation in the Discussion of the original version to which we have added the following note “We believe that GRADE scoring and recommendations are a next-step into the implementation of a risk-based mitigation of contamination of AS DI and HH products.” see in Revised Version Track Changes Page 30, Lines 828 - 830.
Point 6: Comment 7. In tables 2B and 3B, the term "H2O2" should be used instead of "hydrogen peroxide, HP" to be consistent with all of the other generic names.
Response 6: We thank the reviewer for this suggestion but do not fully understand the question. Tables 2B and 3A currently mention “H2O2” in the header, with the abbreviation spelled out as in the Table legend. Did the reviewer might have meant the opposite, i.e. using the generic name “hydrogen peroxide” instead of the formula? Thank you for clarification.
Point 7: Comment 8. The subsection of 4.8 (Internal and external validity) have a different format, the subheading of the section is written in bold letters at the beginning.
Response 7: We thank the reviewer for his/her remark, we have turned the bold letters to the regular edits.
Point 8: Comment 9. The author's clarity in stating "(6) Outstanding issues and research topics and (7) Limits and Strengths" is remarkable. However, the conclusion section of this manuscript is quite vague and lengthy. Some conclusions should state the primary point of the obtained major content and make recommendations for clinical practice. Additionally, this research's conclusion section requires a suggestion for further research.
Response 8: We thank the reviewer for the comment on the conclusion. We have revised the Conclusion to make it more specific and to add the future research questions. The adapted text is as follows (Revised Version Track Changes, Page 30 Lines 847 - 851): “Although underreported, contaminated AS, DI and HH products may act as reservoirs of healthcare-associated outbreaks in LMIC with substantial case fatality and associated with specific risk factors. Domains of implementation research are field-adapted containers and reprocessing, in-country production, water quality, biofilm control, healthcare providers’ practices and the role of bar soap.”
Point 9: Comment 10. Finally, I am grateful for the chance to review this submission. Although the topic is valuable and interesting, further interpretation and motivation is required. Wishing the authors success for their paper publication, I hope the suggestions are beneficial.
Response 9: We thank the reviewer and appreciate the extensive review of our manuscript.

Author Response
Reply to comments of Reviewer 4:
This is a comprehensive and detailed systematic review of healthcare-associated outbreaks and cross-sectional surveys on contamination of antiseptics and disinfectants in healthcare settings in low- and middle-income countries. It provides a survey from 1977 to the presence. Products most associated with outbreaks and contaminations, contaminating bacteria, risk factors and outstanding issues are presented and discussed as well as potential bias and validity issues. A very useful list of “Best Practices” to avoid contamination in the future is presented. The authors have done enormous work, which leads to respective important conclusions and confirms the importance of hygiene in medicine.
Minor specific comments:
Point 1: Page 2, line 50: Please explain “low-level” disinfectants.
Response 1: We thank the reviewer for his/her comment
Low level disinfectants are disinfectants which destroys vegetative bacteria and some fungi and viruses but not mycobacteria or spores; they are used for disinfection of noncritical patient care items or surfaces (in the absence of visible blood) (Rutala2019 Am J Infect Control 2019; 47: A3- A9; CDC2019 Best Practices for Environmental Cleaning in Healthcare Facilities: in Resource-Limited Settings) (References 15 and 16 in the manuscript). To make it clear to the reader, we rephrased the sentence as follows (Revised Version Track Changes, Page 2, Lines 87 - 89): “Environmental cleaning and decontamination processes – for which low-level disinfectants are used [15, 16] - reduce the bacterial load in the hospital environment and figure among the IPC core components as well [4]. Further, we added examples of low-level disinfectants on page 2, lines 89-91.”
Point 2: Page 3, lines 98-99: Please expand the definition of pseudo-outbreaks. In the present form, it may be unclear.
Response 2: We thank the reviewer for this comment. The definition cited is as follows: “Pseudo-outbreaks were defined as contamination of clinical specimens in the absence of exposure of the patient [16].”, with reference 16 referring to Curran, E. T., Outbreak Column 7: Pseudo-outbreaks (part 1). J. Infect. Prev. 14 (2013) 69–74. To make the term more explicit and to distinguish it from its other significance (i.e. an epidemic increase in numbers of cases which is in fact not a real increase), we replaced the definition as currently written by the more extended definition we used in a previous manuscript (cited as reference 9) and we added an example: “The term pseudo-outbreak refers to false-positive cultures of clinical specimens caused by contaminated products in the absence of patient colonization, infection, and exposure [18, 19], such as in the case of blood culture contamination induced by contaminated antiseptics used for wiping the stopper of the blood culture bottles before inoculation [9]” (Revised Version Track Changes, Page 3, Lines 110 - 114).
Point 3: Page 10, line 6 from above:
Dakin-Cooper solution presently marketed is 0.5g sodium hypochlorite / 100 ml water in general. Was it a chloramine T solution in the mentioned publication or hypochlorite? Please clarify. In both cases, I do not understand the expression “slow releasing” chlorine product. NaOCl and chloramine T do not “slowly” release the active chlorine.
Response 3: In the article (reference 49, Cissé M, Médecine et Maladies Infectieuses 1987: 5; 260-3) dated from 1987 and originated from Dakar, Senegal, It mentions “solution of Dakin antiseptic” (without further details), in which intravenous catheters were soaked. We corrected and replaced “Dakin-Cooper” by “Dakin solution” The revised sentence is (Revised Version Track Changes Page 10, Lines 315: “… and a Dakin solution (i.e. a stabilized chlorine product used as antiseptic”). We also corrected and replaced “Dakin-Copper solution” by “Dakin solution” at Pages 13, Line 493 and Pages 18, Line 630 respectively and made the same correction in Supplementary Table 2.
Point 4: Just a comment to the neutralizer issue (pages 12-13): Active chlorine compounds are neutralized on agar plates like trypic soy within approximately 3 min so that usage of neutralizer probably would not have changed the detection rates. With tensidic compounds, it may be different and actually obligatory to use neutralizers. [In killing assays, particularly at low incubation times, neutralizers or respective dilutions (if no neutralizer exists for a compound) are in any case obligatory, of course.]
Response 4: We thank the reviewer for this comment. As to the present articles, most articles did not provide complete details about the culture media used and if done, most descriptions used general terms (e.g., nutrient agar, blood agar) without stating the base (like tryptic soy agar). So, we take note of the reviewer’s valuable comment, but for the moment did not explore how to integrate it into the manuscript.
Point 5: Page 10, lines 24-25 (comment): This statement is very important. Generalizing the statement, the reason for contamination is actually not acquired resistance to antiseptics, but mistakes in the daily practice. Maybe, it can be repeated at important position in the abstract or conclusions.
Response 5: We fully agree with the generalized statement that low concentrations/high dilutions, water-based products and numerous environmental (water quality, biofilm) and human factors (human resources, practices of prolonged in-use, reprocessing of containers, errors in preparation, topping-up) are far more contributing to contamination than does acquired resistance to antiseptics. However, in the absence of data of acquired resistance in the articles retrieved, we feel it is not possible to make an evidence-based statement about it. We believe however such a statement should fit in an editorial or comment on the Special Issue of Hygiene “Emerging Biocide Resistance – Frequency, Drivers, Relevant Outcomes and Containment Strategies”.
Point 6: It is a pity that microorganisms grown and identified during outbreaks and suspected for contamination of antiseptics / disinfectants are generally not tested specifically in quantitative killing assays against use concentrations of these substances. Very probably, only bacterial spores and aggregate forms like biofilms would survive alcohols and active chlorine compounds. I am unsure if this is also valid for tensidic compounds at high application concentration of the antiseptic (at low concentrations, resistance against tensidic compounds is already proven).
Response 6: We fully agree with the reviewer, we refer also to our reply to his/her comment below.
Point 7: Page 13, line 105 (comment): Also in this study, it would have clarified the just mentioned issue if the isolated Burkholderia had been tested directly in 70% alcohol. I am rather sure that it would have been killed rapidly.
Response 7: Here we also agree with the reviewer. The authors of this paper (Nasser et al. Infect. Control Hosp. Epidemiol. 2004; 25: 231-239) described a protracted outbreak of B. cepacia infection in war-torn Lebanon. They did not perform quantitative killing assays but referred to the nutritional versatility of B. cepacia as a (partial) explanation for the contamination of ethanol 70%. To our opinion and experience, given the setting, a too low concentration of the procured ethanol would be a first-to-exclude factor. In a complimentary scoping review about contaminated AS DI and HH products submitted (see comment below, reference no. 9), we advocate for availability of simple methods for verification of product concentrations.
Point 8: Page 14, line 3: S. aureus are not rods. Please correct.
Response 8: We thank the reviewer for this remark. Indeed, Staphylococcus aureus is cocci instead of rods. We made the correction adequately. The adaptation is as follows “… the remaining species were Gram-positive cocci and rods including Staphylococcus aureus and Bacillus spp., respectively" (Page 14, Line 503 – 504).
Point 9: Page 31, reference no. 9. What does “Unpublish (2023)” mean? This is an important reference, and its status should be clear. Please clarify.
Response 9: Reference no. 9 refers to a manuscript entitled “Bacterial contamination of antiseptics, disinfectants and hand hygiene products in healthcare facilities in high-income countries: a scoping review” of which I am lead author too and which has equally been submitted to Special Issue of Hygiene “Emerging Biocide Resistance – Frequency, Drivers, Relevant Outcomes and Containment Strategies Hygiene”. We have currently submitted its revised version and – providing acceptance of the revised manuscript - I will add the full publication details to the reference list of the present manuscript. In the revised version of the manuscript, we updated this reference as follows “… submitted” (Page 31, Reference 9, Line 904).

Round 2
Reviewer 2 Report
Dear
The manuscript is approved and can be published in this format.
Best,
Reviewer 3 Report
The author has satisfactorily and totally edited. This revision also has significantly improved the manuscript.